# Re-convolving the compositional landscape of primary and recurrent glioblastoma reveals prognostic and targetable tissue states

Osama Al-Dalahmah [1,2,6] ✉, Michael G. Argenziano[3,6], Adithya Kannan [3,6], Aayushi Mahajan[3], Julia Furnari[3], Fahad Paryani[4], Deborah Boyett[3], Akshay Save[3], Nelson Humala[2,3], Fatima Khan[1], Juncheng Li[1], Hong Lu[1], Yu Sun[1], John F. Tuddenham [5], Alexander R. Goldberg [1], Athanassios Dovas[1], Matei A. Banu[3], Tejaswi Sudhakar[3], Erin Bush[4], Andrew B. Lassman[2,5], Guy M. McKhann[2,3], Brian J. A. Gill[2,3], Brett Youngerman [2,3], Michael B. Sisti[2,3], Jeffrey N. Bruce [2,3], Peter A. Sims[2,5], Vilas Menon [4] ✉ & Peter Canoll [1,2] ✉

Glioblastoma (GBM) diffusely infiltrates the brain and intermingles with non-neoplastic brain cells, including astrocytes, neurons and microglia/myeloid cells. This complex mixture of cell types forms the biological context for therapeutic response and tumor recurrence. We used single-nucleus RNA sequencing and spatial transcriptomics to determine the cellular composition and transcriptional states in primary and recurrent glioma and identified three compositional 'tissue-states' defined by cohabitation patterns between specific subpopulations of neoplastic and non-neoplastic brain cells. These tissue-states correlated with radiographic, histopathologic, and prognostic features and were enriched in distinct metabolic pathways. Fatty acid biosynthesis was enriched in the tissue-state defined by the cohabitation of astrocyte-like/mesenchymal glioma cells, reactive astrocytes, and macrophages, and was associated with recurrent GBM and shorter survival. Treating acute slices of GBM with a fatty acid synthesis inhibitor depleted the transcriptional signature of this pernicious tissue-state. These findings point to therapies that target interdependencies in the GBM microenvironment.

Glioblastoma (GBM) is the most malignant glial tumor of the brain and is refractory to current treatment. Although gross surgical resection of the visible tumor is sometimes feasible, glioma cells infiltrate the brain beyond the resection margins. While many studies have characterized the transcriptional and genomic features of GBM cells and glioma-associated microglia/myeloid cells, a comprehensive analysis of other cells in the GBM microenvironment, and the patterns of cohabitation of different cell types is lacking. Previous studies have shown that the composition of glioma-infiltrated samples varies from cellular tumor comprised of GBM and myeloid cells, to minimally infiltrated GBM margin tissue composed largely of non-neoplastic brain microenvironment cells, including neurons and glia[1-3]. This is the microenvironment into which tumor cells migrate and proliferate, leading to recurrence, and is also the target of adjuvant therapy. Thus, understanding the cellular milieu of the tumor microenvironment at presentation and recurrence, including both neoplastic and non-

neoplastic cells, is vital for advancing the management of GBM. Our goal is to determine patterns of cellular composition and transcriptional states in primary and recurrent GBM, including both neoplastic glioma cells and non-neoplastic brain cells.

Early studies used bulk RNA-sequencing approaches to understand GBM states in MRI-localized samples from contrast-enhancing (CE) and non-contrast-enhancing (NCE) margins[3–6]. Higher resolution is attained using single-cell RNAseq (scRNAseq) approaches, which are being increasingly used to understand heterogeneity in gliomas. Several studies have employed scRNAseq from freshly resected surgical samples to explore the heterogeneity of GBM[2,7–12]. These studies have significantly advanced our understanding of the heterogeneity and pathology of glioma. However, application of whole-cell scRNAseq is faced with practical challenges related to the limitations of acquiring and processing freshly resected glioma tissue and the technical incompatibility with banked frozen glioma tissue. Moreover, scRNAseq is limited in sampling non-neoplastic cells of the microenvironment like neurons and astrocytes, which are major constituents of the tumor-margins[2,8–11], in part because of cell-type survivability/selection bias during tissue dissociation. Thus, while advances have been made in defining the genetic alterations in glioma[13,14] and the transcriptional states of glioma cells and immune cells[15–18], comprehensive analyses of cellular composition and diversity of cellular phenotypes in primary and recurrent gliomas remain a challenge.

In this work, we circumvent these limitations of scRNAseq by using single-nucleus RNA-sequencing (snRNAseq), allowing us to analyze frozen tissue, and inclusively sample cells of the microenvironment from primary and recurrent glioma. We sample glioma-infiltrated tissue from cellular tumor to minimally infiltrated surrounding brain tissue at the single-cell level. Transcriptional analysis of copy-number variations (CNVs) provided a metric to distinguish neoplastic (CNVpos) and non-neoplastic (CNVneg) nuclei, and unbiased clustering reveals that primary and recurrent tumors harbor CNVpos glioma cells with similar transcriptional states. Conversely, the microenvironment of primary and recurrent glioma displays distinct cell-type-specific states and different compositional landscapes. Leveraging information from the snRNAseq-derived compositional make-up of glioma-infiltrated samples defines three generalizable "tissue-states" with each tissue-state showing enrichment for specific gene signatures that can be identified in bulk RNAseq samples. We also examine these compositional patterns using spatial transcriptomics, which reveals colocalization of specific neoplastic and non-neoplastic cell types. We demonstrate that tissue-states are prognostically relevant and display metabolic dependencies that can be pharmacologically targeted.

## Results

### Transcriptional analysis of the glioma microenvironment reveals prognostically significant subpopulations of non-neoplastic astrocytes

Given the importance of glioma microenvironment in tumor progression, we decided to investigate the implications of microenvironmental states on the prognosis of GBM. To achieve this, we first identified neoplastic and non-neoplastic nuclei based on chromosomal copy-number variation (CNV) inference (Supplementary results). Based on the repertoire of transcriptional states of glioma cells that have been previously described[2,7–10,12], we confirmed that our CNV-positive (CNVpos) neoplastic nuclei from primary and post-treatment recurrence GBM recapitulate known transcriptional states. We provide this data in the supplementary results including discussion of glioma states in primary and recurrent glioma (Supplementary Figs. 1, 3), CNV analysis of primary and recurrent glioma samples (Supplementary Figs. 2, 4), localization studies of glioma states in the tissue (Supplementary Fig. 5), and details on other low-grade glioma and epilepsy

samples included in this study (Supplementary Figs. 6, 7). We focused on the non-neoplastic CNV-negative (CNVneg) nuclei of the glioma microenvironment and combined in our analysis nuclei from primary and recurrent glioma, as well as nuclei from low-grade glioma (LGG) and epilepsy, to include a spectrum of neurological diseases with alterations to non-neoplastic cells in the brain microenvironment. The clinical data on the samples, QC metrics, and number of nuclei per lineage/cluster is provided in Supplementary Dataset 1. Our CNVneg nuclei datasets included 16831 nuclei: 6929 from primary glioma, 6008 from post-treatment recurrent glioma, 2875 from epilepsy, and 1019 from LGG. We projected these nuclei in UMAP space and assigned cell lineages as shown in Fig. 1a. The expression of a select number of marker genes per lineage is shown in Fig. 1b. We present the results on myeloid lineage nuclei in the supplementary results (Supplementary Fig. 9), which demonstrates that monocyte-derived tumor-associated macrophages (TAMs) were enriched in recurrent glioma, while microglia-derived TAMs were enriched in primary glioma, consistent with a previous report[15].

We focused on astrocytes, which are key elements of the glioma microenvironment and are not well represented in glioma single-cell RNAseq datasets[2,7–12,19]. A recent paper implicated GBM-associated astrocytes in promoting an immunosuppressive microenvironment[20]. Moreover, the distinction between tumor-astrocytes and reactive astrocytes is of major diagnostic importance in neuropathology. Thus, we analyzed astrocytes (707 nuclei−284 from primary glioma, 254 from recurrent glioma, 45 from LGG, and 121 from epilepsy) in isolation from other cell types, performed linear dimensionality reduction, and clustered them into three states; Ast1−protoplasmic astrocytes, Ast2−reactive astrocytes with expression of oligodendroglial and neuronal genes, and Ast3−reactive astrocytes with inflammatory gene expression (Fig. 2a, and Supplementary Dataset 4). The astrocytes are projected by disease condition in Fig. 2b. Clustering of astrocytes was based on the enrichment of three genesets with pre-defined genes relevant to astrocyte function (Supplementary Dataset 4 and Fig. 2c, d). Expression of select markers of these astrocytes states (clusters) is shown in Fig. 2c. Since astrocytes and glioma shared gene signatures (for example, CLU and LGALS3 expression), we performed differential gene expression analysis between primary and recurrent glioma non-neoplastic astrocytes and all CNVpos glioma nuclei and identified 1620 genes that were higher in astrocytes compared to glioma and 3380 that were higher in glioma compared to astrocytes. Examples of genes higher in non-neoplastic astrocyte include genes associated with Alzheimer's disease (CLU, APOE)[21,22], metallothionein genes (MT1H, MT1G, MT1M, MT1F, MT1E, MT1X, MT2A, and MT3−increased in reactive astrocytes[23]), Synuclein genes (SNCA, SNCB, and SNCG), WIF1, CHI3L2 (associated with poor prognosis in glioma[24]), ALDOC, ALDOA, AQP4, carbonic anhydrases CA2 and CA11, and CXCL14, a cytokine implicated in promoting glioma invasion[25] (Supplementary Dataset 4). Conversely, genes higher in CNVpos glioma include EGFR, PTPRZ1, NOVA1, CD24, Nestin (NES), SOX5, and SOX4. We used KEGG pathway enrichment analysis to query the function of these genes (Fig. 3). Further analysis of the differentially expressed genes showed that several KEGG pathways were enriched in genes higher in non-neoplastic astrocytes (Fig. 3a), with some relating to neurodegeneration such as Parkinson disease, and prion disease. Notably, these signatures are highly enriched in oxidative phosphorylation genes (Supplementary Dataset 4), which are dysregulated in neurodegenerative diseases[26]. Moreover, other metabolic pathways enriched in astrocyte DEGs included metabolism of fatty acids, glycolysis, TCA cycle, and ferroptosis. Conversely, KEGG pathways increased in CNVpos tumor-astrocytes were largely related to DNA replication, cancer-related pathways including ErbB and MAPK signaling, DNA replication and mismatch repair (Fig. 3b and Supplementary Dataset 4).

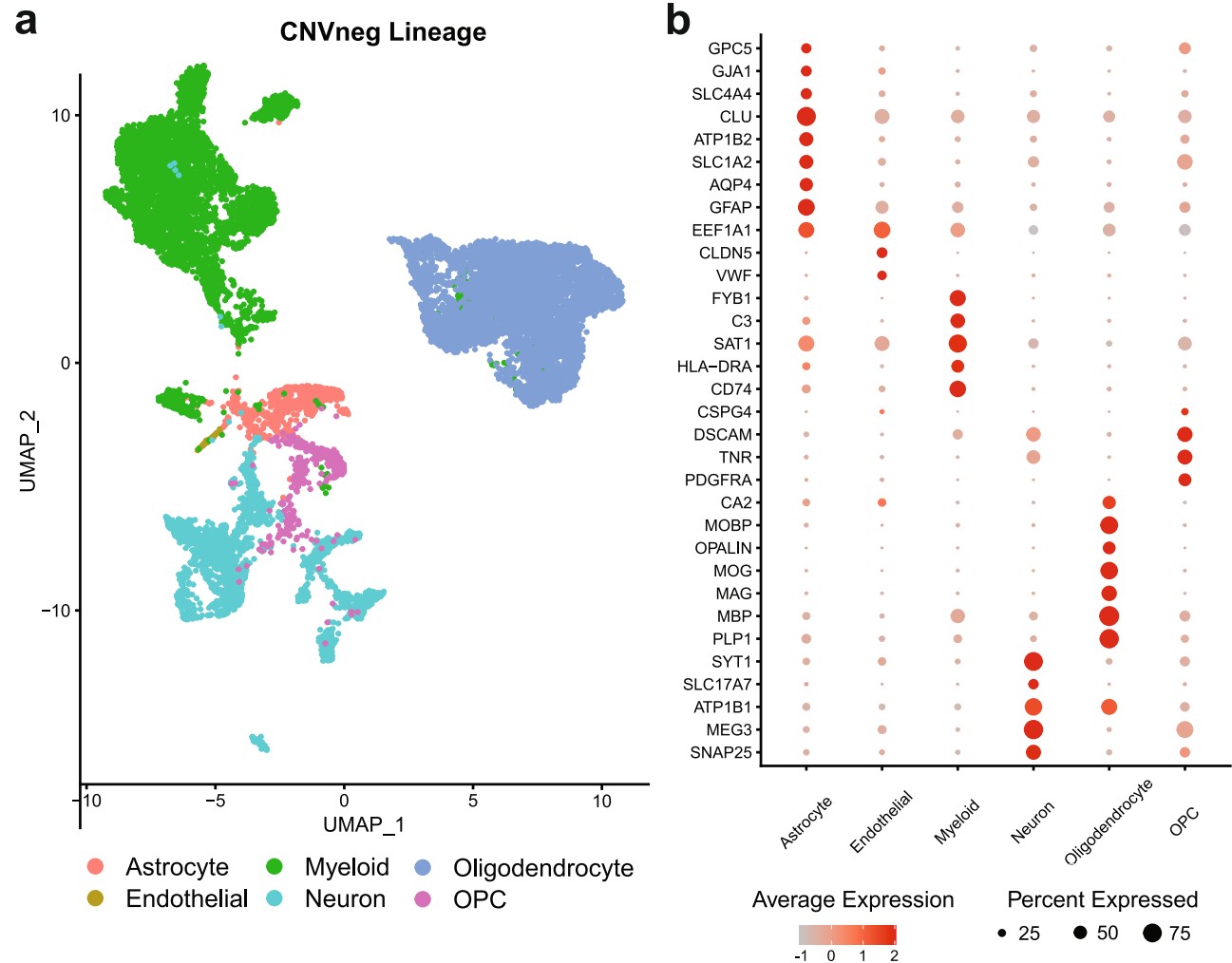

**Fig. 1 | snRNAseq identifies non-neoplastic nuclei in the tumor microenvironment. a** Uniform-manifold approximation and projection (UMAP) graphs showing putative non-neoplastic (CNVneg) nuclei from primary glioma, recurrent glioma, low-grade glioma (LGG)—and epilepsy (see supplementary data for the analysis of LGG and epilepsy cases). The nuclei are color-coded by lineage (oligodendrocytes, oligodendrocyte-precursor cells (OPC), neurons, astrocytes, myeloid cells, and endothelial cells). **b** Dot plots showing normalized expression of select lineage genes (rows) in the lineages from a (columns). The size of each circle corresponds to the proportion of each lineage that expresses a given gene.

## snRNAseq and spatial transcriptomics reveals patterns of cohabitation between neoplastic and non-neoplastic cell types

Given the heterogeneity of cellular states of glioma and non-neoplastic cells in the glioma microenvironment, we hypothesized that the transcriptional landscape of GBM is determined by patterns of cohabitation of specific types and transcriptional states of neoplastic and non-neoplastic cells. To test this hypothesis, we first asked if specific glioma, or brain microenvironment lineages were differentially abundant or depleted across primary and recurrent glioma using a regression model[27] to test for differential abundance (Fig. 4a). The results showed that for CNVpos cells, gl_Mes2 were significantly more abundant in recurrent glioma, while gl_PN1 were more abundant in primary glioma, (Benjamini–Hochberg adjusted p-values (q-value) 3.99e-2 and 1.318e-5, respectively). For the CNVneg cells in the glioma microenvironment, OPCs were significantly more abundant in primary glioma (q-value 1.085e-03). These results show that patterns of cellular composition vary in primary and recurrent glioma, and likely contribute to determining the transcriptional landscape of glioma.

To uncover patterns of 'tissue-states' with correlated cell states/lineages, we took advantage of the relatively unbiased sampling of cellular composition in the brain tumor microenvironment provided by snRNAseq. We approximated the cellular composition of each surgical sample by recombining the cells from all the distinct cell populations, as identified by snRNAseq, to create a compositional matrix containing the abundance of all cell types across all samples (Supplementary Dataset 1). The cellular composition matrix includes three astrocytic clusters (Ast1-3), five immune-cell states (Myel1, moTAM, mgTAM, prTAM, and T cells—see supplementary results and methods), neurons, oligodendrocytes, endothelial cells, OPCs, and glioma cells. We then used principal component analysis of the resulting cellular composition matrix and identified the compositional features that account for the variance across the samples (Fig. 4b). We used the glioma states as supplementary quantitative variables[28]—the coordinates of which can be predicted from the other variables input into the PCA analysis. The results showed that the relative abundance of CNVpos glioma cells vs. CNVneg non-neoplastic cells (neurons, oligodendrocytes, OPCs) is the major feature of the first principal component, and the abundance of reactive astrocytes (Ast3), macrophage-like myeloid cells (moTAM), and T cells is the major feature of the second principal component. Notably, the abundance of a specific subpopulation of astrocyte-like/mesenchymal glioma cells (gl_Mes2) was also highly correlated with the second principal component (PC2). These findings indicate that specific subpopulations of neoplastic and non-neoplastic cells tend to co-inhabit glioma samples.

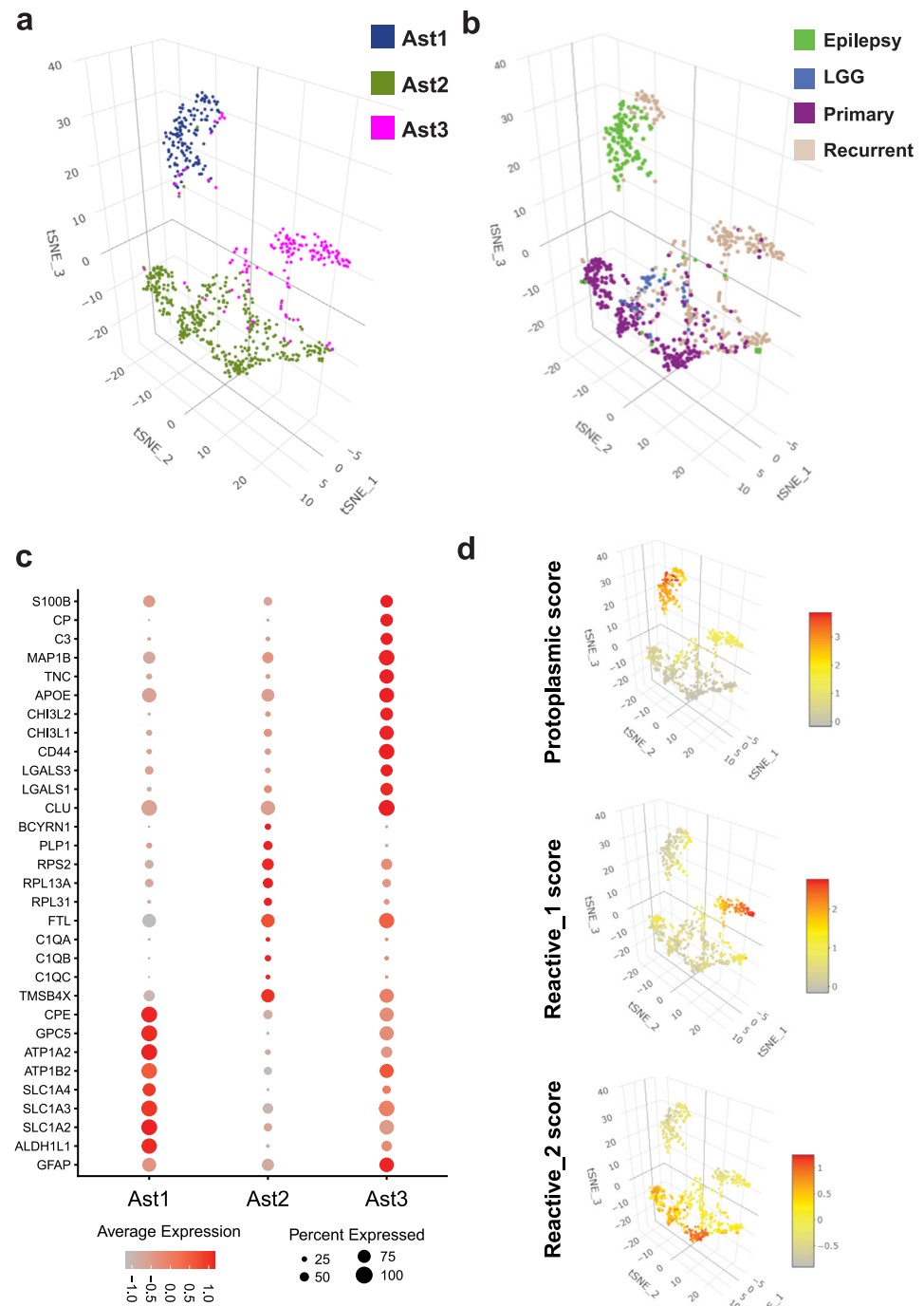

**Fig. 2 | snRNAseq identifies three transcriptionally distinct astrocytes states in the glioma microenvironment. a** Three-dimensional tSNE plots showing all astrocyte nuclei color-coded by astrocyte state (Ast1−protoplasmic astrocytes, Ast2−reactive astrocytes with misexpression of non-astrocyte lineage genes, and Ast3−reactive astrocytes with expression of inflammatory genes. **b** Three-dimensional tSNE plots showing all astrocyte nuclei color-coded by disease condition. **c** Gene expression dot plots showing select gene marker expression for the astrocyte states. **d** tSNE plots showing the enrichment of gene signatures used for astrocyte clustering in astrocyte nuclei.

To assess if the cohabitation of cell types and transcriptional states is prognostically relevant, we used the IDH-WT GBM TCGA and CGGA survival datasets and performed a log-rank test on samples with positive vs. negative PC2 signature enrichment and found that positive enrichment is significantly associated with poor survival (Fig. 4c). These data show that glioma-infiltrated tissue shows patterns of cellular composition driven by cohabitation of specific cell-types and transcriptional states and reveal prognostically relevant gene signatures that span across both neoplastic and non-neoplastic cell types.

To further characterize the cohabitation of specific cell types and transcriptional states in GBM, we analyzed nine samples of IDH-WT GBM infiltrated brain tissue using spatial transcriptomics (ST-Supplementary Dataset 1, Fig. 5a, b and Supplementary Figs. 10, 11). We deconvolved the ST data using RCTD[29] and analyzed the spatial relationships between cell types. To improve the accuracy of deconvolution results, we incorporated snRNAseq from the same tissue samples used to generate the ST data when possible (Validation snRNAseq dataset−see supplementary information and Supplementary Fig. 8).

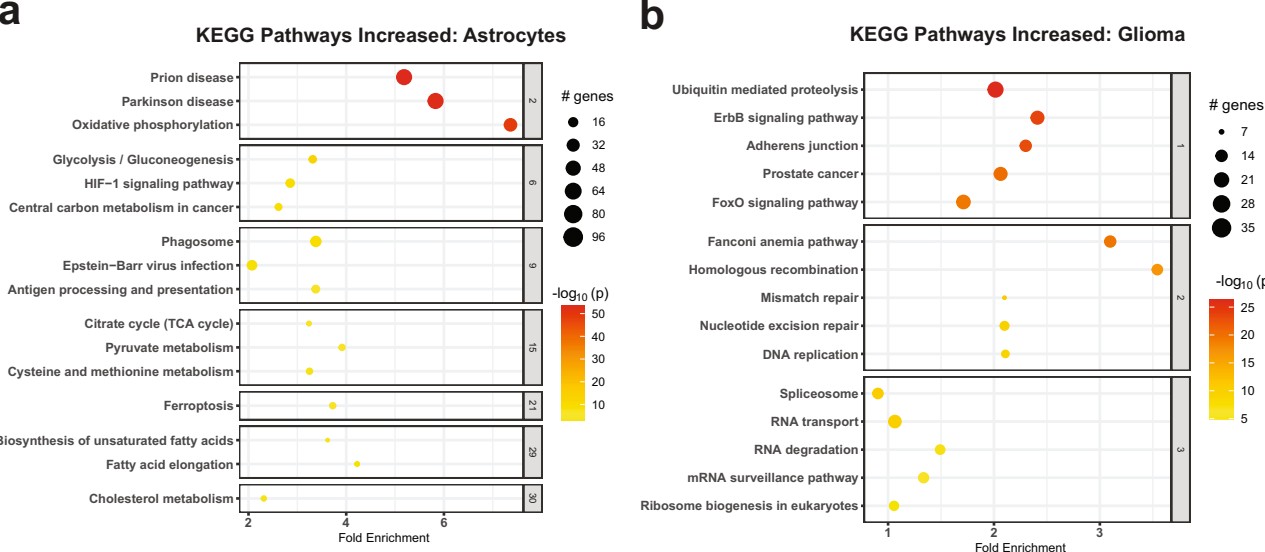

**Fig. 3 | Differentially expressed genes in astrocytes and glioma are enriched for different gene ontologies. a, b** Active subnetwork enrichment analysis of KEGG pathways in genes differentially expressed in CNVneg glioma-associated astrocytes compared to CNVpos glioma cells in primary and recurrent IDH-WT glioma. Fold-enrichment is represented on the *x*-axis and the pathways in the *y*-axis. The pathways are clustered to denote shared genes driving enrichment. Significance was calculated using a Fisher exact test. The size of the circle per pathway denotes the number of enriched genes, and the negative log10 of the Bonferroni adjusted *p*-value is represented by color. Pathways enriched in genes significantly higher in astrocytes compared to glioma cells are shown in **a** and include neurodegenerative diseases and oxidative phosphorylation, metabolism including fatty acid metabolism. Pathways enriched in genes significantly higher in glioma cells compared to astrocytes are shown in **b** and include DNA replication, splicing, and ErbB signaling.

We determined the proportion of our 18 cell types present in each of the 9017 transcriptomic capture spots that comprised our ST experiments and quantified the relationship between different cell types using spatial cross-correlation. Spatial cross-correlation quantifies the correlation between the proportion of cell type A in any given spot and the proportion of cell type B in that spot's neighbors. By evaluating this metric for every spot, between all pairwise comparisons of cell types, we were able to quantitatively assess the geographic relationship between cell types and determine which global patterns of cohabitation were statistically significant (Fig. 5b). Clustering of the spatial cross-correlations showed three main clusters: 1—showing positive and statistically significant cross-correlations between non-neoplastic cell types such as neurons, oligodendrocytes, non-reactive astrocytes (Ast1, Ast2), and OPCs; 2—showing statistically significant spatial correlations between gl_Mes2, reactive astrocytes (Ast3), moTAM, and T-cells; and 3—showing statistically significant spatial correlations between endothelial cells and several CNVpos glioma cell subtypes. Interestingly, gl_PN2 and gl_Mes2 were significantly spatially cross-correlated, and our independent validation dataset analyzed by RNA-scope for gl_Mes2 and gl_PN2 markers (Supplementary Fig. 5) shows both are significantly more abundant in cortical regions. These findings provide additional evidence to support cohabitation of these cell types.

**Re-convolution of snRNAseq identifies three tissue states based on cellular composition of glioma and its microenvironment**
Driven by the above findings, we clustered the snRNAseq samples from our discovery dataset into 3 distinct "tissue-states" based on the approximated cellular compositions described above; tissue-state A samples are predominantly composed of non-neoplastic brain cells, including neurons oligodendrocytes, and OPCs, tissue-state B samples are enriched in reactive astrocytes, myeloid/macrophages, and T-cells, and tissue-state C samples are predominantly composed of CNVpos glioma cells (Fig. 6a, b). Based on the results of our compositional clusters/tissue states, we are able to assign tissue states to the external

validation set based on *k*-means classification (validation set—Supplementary Fig. 8h). To generate a gene signature for each tissue state, we combined the snRNAseq for all nuclei in each sample and performed differential gene expression analysis between tissue-state clusters, using the pseudobulk expression profile of each sample as a biological replicate. This analysis identified the top-differentially expressed genes unique to each tissue state (Supplementary Dataset 7). To assess the generalizability of the three tissue-states, we performed single-sample GSEA analysis for the tissue state gene signatures using a dataset of bulk RNAseq analysis performed on 91 primary and recurrent MRI-localized samples from 39 patients. We found that these samples separated into 3 compositional clusters based on their enrichment score for snRNAseq-defined "tissue-states" (Fig. 6c). We refer to the compositional clusters and tissue-states interchangeably henceforth. Further analysis revealed that these tissue-state gene signatures are enriched for specific biologically relevant functional ontologies. For example, tissue-state A is enriched for genes involved in synaptic transmission, tissue-state B is enriched for genes associated with inflammation, and tissue state C is associated with cell proliferation (Fig. 6d). These three tissue-states are further demonstrated by projecting the RNA-expression levels for canonical markers of the predominant cell types for each tissue-state in Fig. 6e showing RBFOX3 (neuronal marker) in tissue-state A, CD68 (myeloid marker) in tissue-state B, and MKI67 (proliferation marker) in tissue-state C. SOX2 (a pan-glioma marker) was widely distributed across the samples, indicating variable degrees of tumor infiltration across samples in all three tissue-states (Fig. 6e). To further validate these findings, we quantified total cellularity and the IHC labeling indices SOX2, NeuN, CD68, and Ki67 in 45 recurrent and primary glioma samples (Fig. 6f) and found that total cellularity was highest in cluster C, which also had the highest abundance of SOX2+ and Ki67+ cells, while cluster A had the highest abundance of NeuN+ cells, and Cluster B had the highest abundance of CD68 + cells. While Clusters A and B resemble normal and reactive brain tissue, the SOX2 and Ki67 labeling indices indicate that these clusters comprise samples with variable levels of glioma infiltration.

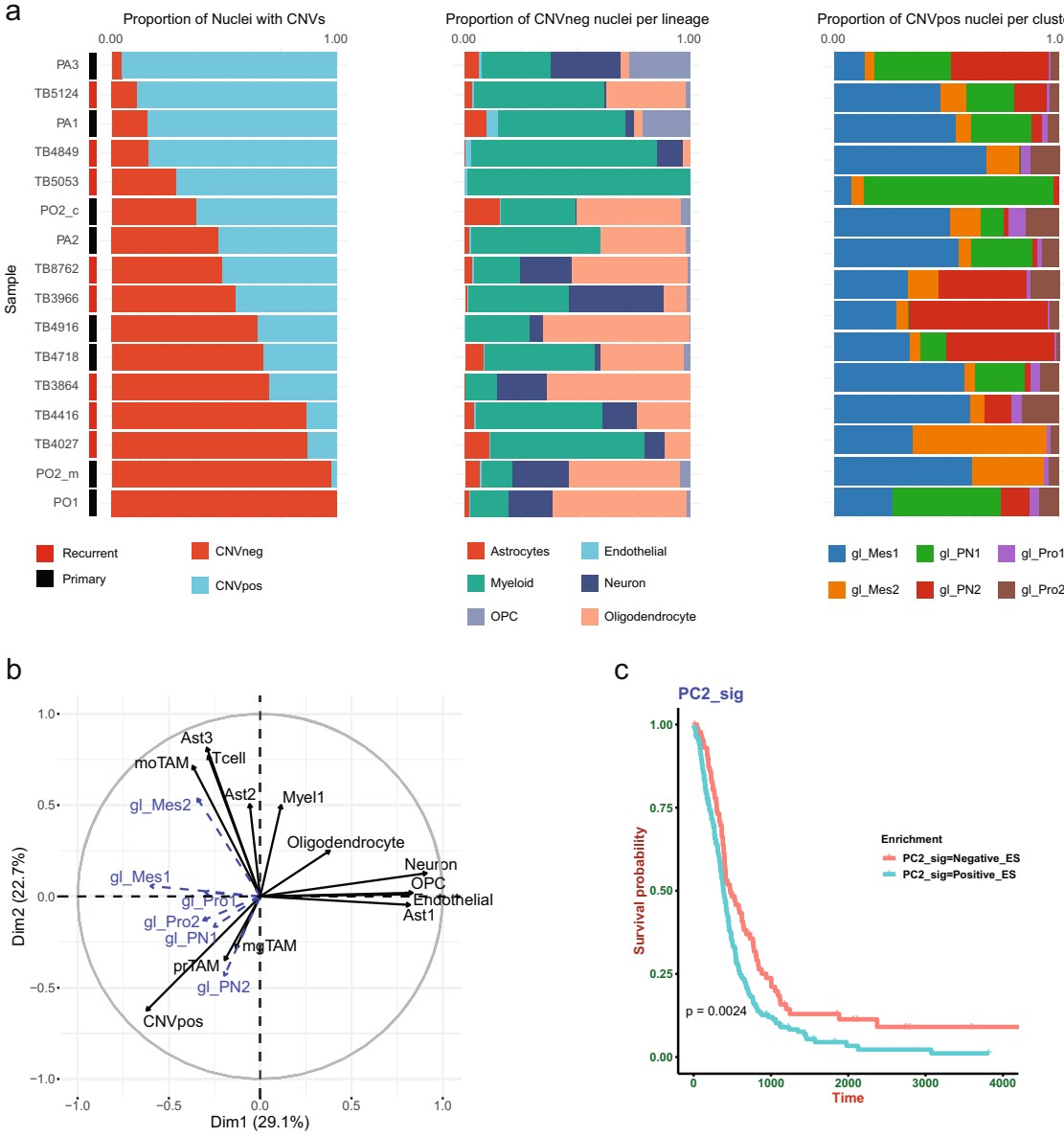

**Fig. 4 | snRNAseq identifies compositional patterns that correlate with survival. a** Bar plots demonstrating the fractional composition of each one of 16 samples analyzed by snRNAseq (8 primary IDH-WT glioma from 7 patients, one case was divided to core and overlying cortex, and 8 recurrent IDH-WT glioblastoma). The left panel shows the fraction of neoplastic (CNVpos) and non-neoplastic (CNVneg) nuclei. The middle panel shows the fraction of the non-neoplastic nuclei contributed by neurons, oligodendrocytes, OPCs, astrocytes, myeloid cells, endothelial cells, and astrocytes. The right panel shows the fraction of the neoplastic nuclei contributed by proneural/progenitor-like glioma (gl_PN1, gl_PN2), astrocyte-like/mesenchymal glioma (gl_Mes1, gl_Mes2), and proliferative glioma (gl_Pro1, and gl_Pro2). The description of glioma states is provided in the supplementary results. **b** Principal component analysis of the fractional composition matrix of 19 samples encompassing eight primary and eight recurrent gliomas plus three epilepsy samples (Supplementary dataset 1). The tissue composition matrix consists of the percentage of nuclei per each tissue state. Immune-cell states are: mgTAMs (microglia-derived Tumor-associated macrophages), moTAM (monocyte-derived TAMs), prTAM (proliferative TAM), Myel1 (baseline myeloid cells), and T cells. Astrocyte states include baseline (protoplasmic) astrocytes (Ast1), reactive CD44 + astrocytes (Ast3), and reactive astrocytes with expression of non-astrocyte genes (Ast2)—see text and supplementary results for additional description of these cell states. CNVpos represents the total percentage of all glioma states per sample. Individual glioma states were not used in PCA calculation, rather they were used as supplementary quantitative variables and their coordinates were predicted from the PCA analysis—see methods. **c** Kaplan–Meier survival plot graphing survival (days) in the combined TCGA and CGGA RNAseq datasets. The samples were classified based on positive or negative enrichment for the PC2 gene signature. Statistical significance was computed using the log-rank test.

To substantiate the clinical relevance of investigating glioma tissue in terms of tissue states, we investigated whether the enrichment of tissue state signatures correlated with survival in the TCGA-CGGA *IDH*-WT glioblastoma dataset. Given that tissue state B was enriched for the gene signatures of Ast3, moTAM, and T-cells (Fig. 7a), and considering our findings in Fig. 4b, c, we expected it to be associated with increased risk of death in survival cohorts. As expected, enrichment of Cluster B gene signature in the *IDH*-WT

TCGA and CGGA datasets was associated with a significant increase in the hazard of death in cox proportional hazard regression model, with covariates controlled for including age, sex, and *MGMT* methylation status (Fig. 7b). In contrast, no significant association with survival was seen for the gene signatures of the individual cell types that compose tissue state B, including gl_Mes2, Ast3, T-cells, and moTAMs (Fig. 7b). To further assess the contribution of Ast3 in this relationship, we adjusted the Cox proportional hazard model by

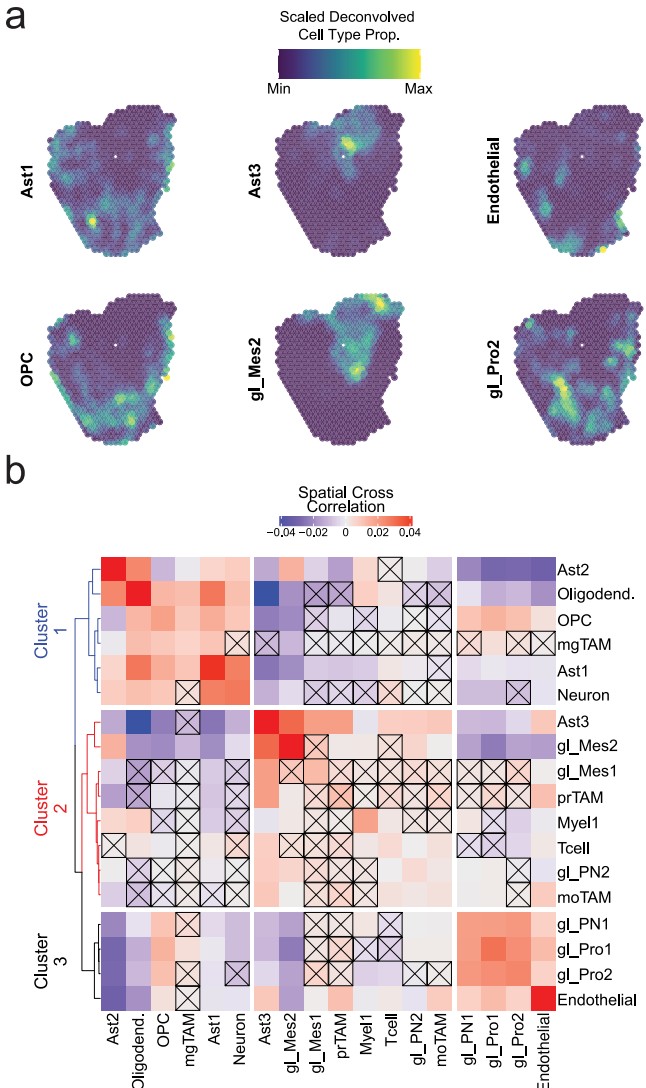

**Fig. 5 | Spatial transcriptomics identifies significant spatial relationships between cell types in GBM. a** Representative plots showing deconvolved proportions of select cell types and glioma states across an ST sample. Each subplot is range scaled for the proportion of that cell type in the sample to show the relative spatial distribution of that cell type. **b** Heatmap showing the average spatial cross-correlation between all cell types at a radius of 900 μm surrounding spatial transcriptomic spots across all nine ST experiments. The diagonal of the matrix, which shows the spatial cross-correlation between a cell type and itself, is an indication of the degree of spatial autocorrelation in that cell type and is not necessarily equal to one. Spatial cross-correlation relationships were tested for significance using permutation (see methods), and non-significant relationships are denoted with an X. Hierarchical clustering of the distance matrix derived from the cross-correlation matrix produced three clusters.

regressing out enrichment for the Ast3 signature and found that tissue state B was no longer significantly associated with an increased hazard of death. To further establish the clinical relevance of taking a tissue state approach in investigating glioblastoma transcriptomics, we asked if tissue state B was differentially enriched in primary vs. recurrent glioblastoma status. This question was especially relevant given that compositional cluster B was largely composed of recurrent glioma samples (Fig. 6a). We thus asked if that signature is positively enriched in RNAseq profiles from previously published paired primary and recurrent glioblastoma samples[30] (Fig. 7c). The results showed significant enrichment in tissue state B signatures in the recurrent GBM samples. Together, the results show

that tissue state B signature is prognostic and enriched during GBM recurrence.

## Glioma-associated tissue states are targetable and associated with distinct metabolic states

Given the distinct cohabitation patterns that drive tissue states, we hypothesized that these patterns of cellular cohabitation are associated with metabolic dependencies. To test this hypothesis, we investigated whether metabolic pathways are differentially enriched in genes differentially expressed between bulk RNAseq samples of the three tissue states. Unbiased analysis of enrichment of KEGG pathways in genes differentially expressed between compositional clusters/tissue-states revealed that they exhibit enrichment of multiple unique and specific pathways (Fig. 8a). Interestingly, several of the tissue state-enriched pathways were metabolic pathways. Tissue-state A showed highest enrichment for neurotransmitter metabolism, oxidative phosphorylation and glutamate metabolism, tissue-state C was most enriched for pyrimidine, folate, and purine metabolism, and tissue-state B showed highest enrichment of fatty acid and lipid metabolism (Fig. 8b). We focused on fatty acid biosynthesis genes, a tissue state B enriched pathway, and projected the average normalized expression per lineage as a heatmap in Fig. 8c. We found that genes in this pathway were distributed across multiple cell types, suggesting that the metabolic status of a tissue can have distinct, but functionally related effects on different cell types in that tissue. Notably, *FASN*, the gene coding for fatty acid synthase (FAS), a rate-limiting enzyme in fatty acid synthesis[31], was most highly expressed in astrocytes and glioma cells (Fig. 8c). FAS inhibition has been shown to kill glioma cells[32], however, the impact of FAS blockade on the glioma microenvironment is yet to be fully explored. Defining the effects of FAS blockade on the glioma microenvironment is important because fatty acid metabolism is a physiologic pathway that involves interactions between multiple cell types that reside in the same habitat. In non-neoplastic brain tissue, fatty acids are synthesized by astrocytes and are distributed to other cells including neurons and oligodendrocytes[31], where they drive physiologic and cellular functions like neuronal maturation, membrane synthesis[33], and neuroprotection[34]. We thus hypothesized blocking fatty acid synthesis pathway would interfere with the cells that make up tissue state B and/or their interactions, and therefore would lead to depletion of tissue state B signature in glioblastoma infiltrated brain. To test this hypothesis, we treated astrocytes and explants of human IDH-WT glioblastoma with the FAS inhibitor Cerulenin (5 mg/ml) and measured gene expression using the PLATE-seq RNAseq (Fig. 9a). Astrocytes treated with Cerulenin exhibited numerous differentially expressed genes compared with DMSO controls (Supplementary Dataset 6, Fig. 9b). Genes increased in treated astrocytes were enriched in KEGG and Reactome pathways involved in mTOR signaling, ferroptosis, and unfolded protein response, while those decreased in treated astrocytes were enriched in pathways involved in cell cycling (Fig. 9c). We then treated IDH-WT glioblastoma explants with DMSO or Cerulenin (Supplementary Dataset 6) and measured gene expression (Fig. 9d). We found that genes increased in Cerulenin treated astrocytes were significantly enriched in Cerulenin treated IDH-WT glioblastoma explants, and that the tissue state B signature was depleted (negatively enriched). It is important to note that the tissue state B gene signature used in this enrichment analysis does not contain any genes that are part of the fatty acid synthesis pathway gene ontology (Supplementary Dataset 7). Overall, these results demonstrate that tissue-states exhibit enrichment of metabolic pathways, which can be targeted leveraging compositional information and metabolic dependencies.

## Discussion

In this work, we investigated the landscape of cellular composition and transcriptional states of neoplastic and non-neoplastic cell types in

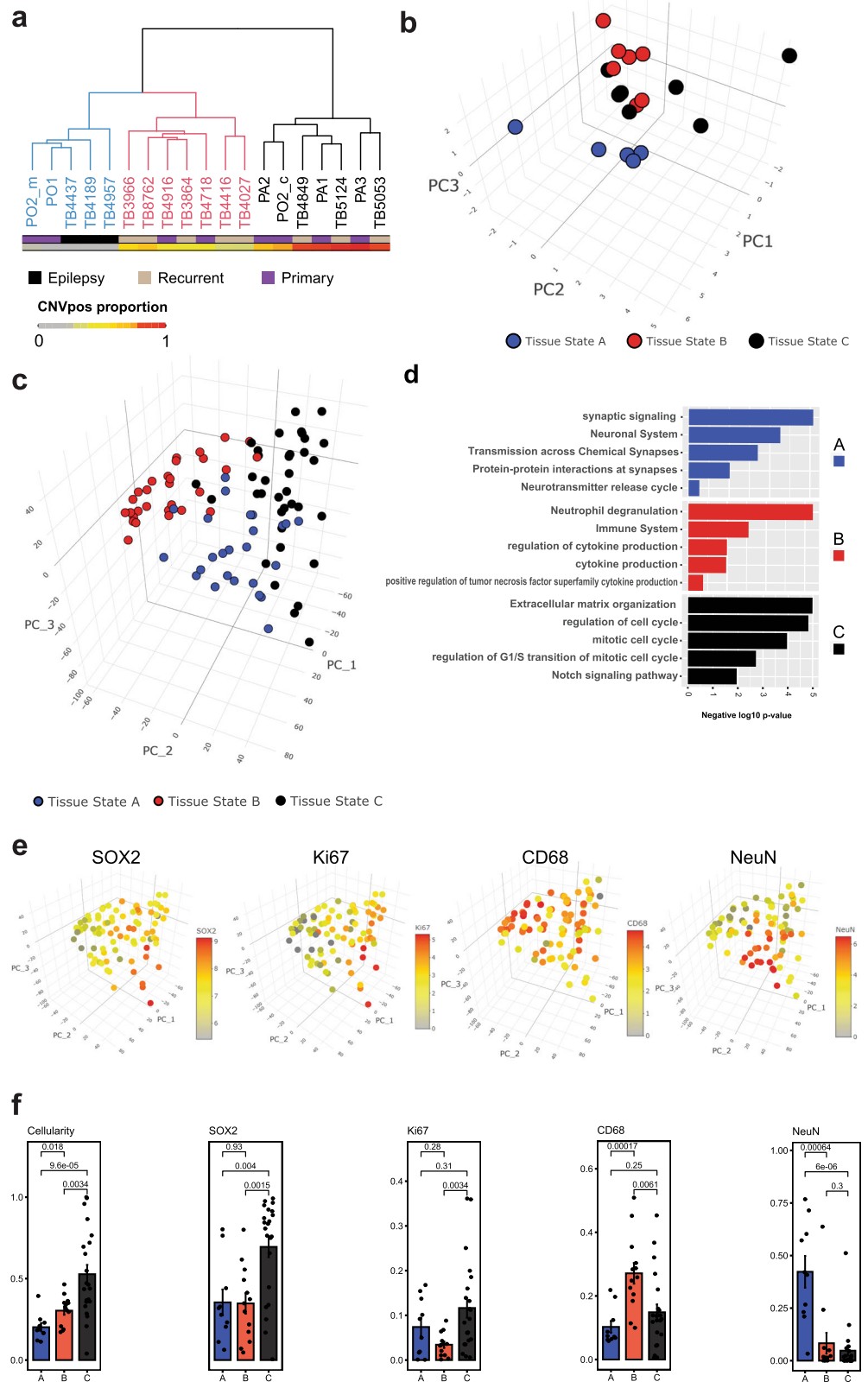

primary and post-treatment recurrent IDH-WT GBM using snRNAseq and spatial transcriptomics. Understanding heterogeneity in GBM is important for guiding treatment and meeting the challenge of recurrence. Recent studies revealed a diversity of glioma states that resemble cell lineages found during development and adulthood[2,9–12,19]. Our study provides a comprehensive analysis of the GBM microenvironment, including non-neoplastic cell types that are sparsely represented in datasets from prior studies using scRNAseq. Using a compositional approach rooted in relatively unbiased sampling of different GBM microenvironment cell types, we discovered that specific cell types/transcriptional states colocalize in "tissue-states". Leveraging insight into correlated cellular states and lineages that co-inhabit tissue samples, we identified gene signatures that classify primary and recurrent GBM tissue into three tissue states: (A)

**Fig. 6 | Tissue composition analysis defines "tissue states" recapitulated in a validation bulk RNAseq dataset. a** Dendrogram of hierarchically clustered glioma and epilepsy samples based on Manhattan sample distance analysis drawn from the fractional composition matrix (see Fig. 4a). Three clusters were identified and are color-coded on the dendrogram in black (Tissue-state C), red (Tissue-state B), and blue (Tissue-state A). The condition (primary, recurrent and epilepsy) and proportion of neoplastic nuclei are indicated. **b** Three-dimensional scatter plot showing the samples in panel **a** projected in the first three principal component loadings—see Fig. 4b for PCA analysis. **c** Bulk RNAseq samples from 91 primary and recurrent IDH-WT glioblastoma samples projected in the principal component space. The samples were clustered (Hierarchical clustering—Ward.D2 method) on the Euclidian distance of the enrichment scores of the genes unique to each tissue state signature into three clusters A–C. **d** Gene ontology term analysis of the differentially expressed genes for each cluster in panel **c**. KEGG, REACTOME, or Biological Process GO pathways are shown in the y-axis. Negative log10 adjusted p-value is shown on the x-axis. **e** Normalized expression of select genes characteristic of each of the clusters projected onto the compositional-signature enrichment score space shown in panel **c**. Red denotes high expression, and gray denotes low expression. NeuN (*RBFOX3*) is highest in the samples of Cluster A. *CD68* is highest in the samples of cluster B. *SOX2* is highest in the samples of clusters B and C. *MKI67* is highest in the samples of cluster C. **f** Quantification of histological cellularity analysis and immunohistochemistry labeling indices of SOX2, KI67, CD68, and NeuN. The labeling index is shown on the y-axis. Note that the y-axis for the cellularity graph is total cellularity normalized to the most cellular sample. The sample clusters are labeled (A–C) as in panel **c**. Indicated p-values were calculated using a Kruskal–Wallis test. $n = 45$ biological samples: 8 for cluster A, 25 for cluster B, and 12 for cluster C. Source Data are provided as a Source data file.

normal brain, (B) reactive/inflammatory tissue, and (C) cellular/proliferative tumor. The tissue states exhibited variable levels of infiltration by glioma cells. We stress that the tissue states we identified do not encompass the entirety of heterogeneity of possible tissue states, and discovery of other tissue states, for example in the context of different treatment scenarios, is highly probable. Also, while snRNAseq provides us insight into cell types that are underrepresented in scRNAseq, snRNAseq exhibits limitations in the ability to identify particular cell states, as has been shown for myeloid cells[35]. Future studies using complementary analysis of scRNAseq and snRNAseq can further elucidate the contribution of particular myeloid activation states to our compositional patterns. That said, the patterns of cohabitation in the tissue state model are further supported by deconvolution of spatial transcriptomics data, which highlights the differential distribution of specific neoplastic and non-neoplastic cell types.

Spatial cross-correlation analysis of our spatial transcriptomics data shows that transcriptionally distinct cell types exhibit significant patterns of colocalization. The patterns we observed using a neighborhood of 900 uM were similar to the tissue state patterns we identified using principal component analysis, demonstrating that these cell composition patterns can be observed using multiple approaches. Analysis of different neighborhood sizes may yield further insight into the mechanisms that drive cohabitation between cell types. Patterns of cohabitation that span large areas are more likely to reflect environmental influences that can impact multiple cell types in a large geographic swathe, such as hypoxia or other metabolic stresses, whereas patterns that vary over smaller distances may reflect the effects of direct cell-cell interactions. Further investigation will be needed to elucidate the exact mechanisms that underlie the organization of cell types into distinct compositional patterns. While we were able to find support for our tissue state hypothesis in our spatial transcriptomics data, additional insight may be provided by improved read coverage and sample preparation. Furthermore, while DAPI provided a measurement of cellularity and NeuN provided information of the distribution of neurons, additional stains, such as standard H&E as well as immunostains GFAP or IBA1, would have provided more insight into the landscape of neuropathologic features in the samples analyzed by spatial transcriptomics.

Importantly, we discovered that enrichment for tissue state B, a reactive state that harbors a reactive astrocyte state (Ast3), was associated with increased risk of death. The presence of tissue-state B was significantly associated with a worse mortality even though its individual component cell types were not, suggesting that the compositional patterns defined by tissue-states contribute to mortality in GBM. We show that gene signatures for these tissue states can also be identified in more accessible bulk RNAseq samples and correlate with immunohistochemical profiles. Significantly, we found that tissues states were transcriptionally enriched in distinct metabolic pathways, and that targeting fatty acid synthesis, a pathway enriched tissue state B, resulted in depletion of that signature in ex vivo GBM slice cultures.

The therapeutic implications of our findings help expand the target of therapy from targeting one gene or one cell type, to targeting tissue states comprising cell populations that co-inhabit the tissue under defined metabolic constraints.

Our analysis of the cellular phenotypes in the glioma microenvironment revealed that subpopulations of non-neoplastic astrocytes show enrichment for abnormal transcriptional signatures that are also seen in the context of neurodegenerative diseases. In contrast to CNVpos neoplastic astrocytes, which express high levels of proliferation and glioma genes, a subpopulation of non-neoplastic astrocytes (Ast3) displayed a reactive signature reminiscent of astrocytes described in neurodegenerative diseases like Huntington disease, Parkinson disease and Alzheimer's disease[23,36]. This phenotype includes enrichment of pathways related to fatty acid metabolism. *CLU*, a gene which codes for clusterin, an astrocyte-expressed apolipoprotein involved in lipid transport[37] and neuroprotection in Alzheimer's disease[21,38], was significantly increased in glioma-associated astrocytes and is a marker of the reactive Ast3. We found that Ast3-like *CLU*-overexpressing astrocytes alter the transcriptional phenotype of glioma in vitro (Supplementary Results and Supplementary Fig. 13), including upregulation of genes involved in glial differentiation and notch signaling. Thus, our results point to commonalities in astrocyte dysregulation across neurologic diseases, which may offer therapeutic targets to be exploited in different clinical scenarios. Future studies are needed to further evaluate the potential of targeting reactive astrocytes as a therapeutic strategy to block GBM progression.

One of the main findings highlighted by our analysis of cellular composition is that specific cell types are correlated with each other both compositionally and spatially, indicating that they co-inhabit the same tissue-states. Cohabitation between cell types and transcriptional states was reflected in enrichment of distinct metabolic pathways. For example, tissue state B was enriched in genes associated with oxidative stress, which determines a cell's sensitivity to ferroptosis-inducing drugs[39], and in fatty acid metabolism, which has been implicated in glioma survival, stemness and progression[32,40,41]. We found that fatty acid metabolism genes were distributed among different cell types in the brain, however, *FASN*, the gene associated with the rate-limiting enzyme in fatty acid synthesis[31] was most highly expressed in astrocytes and glioma cells. Astrocytes play key roles in lipid metabolism; for example, in synthesizing fatty acids necessary for neuronal membranes[33] and catabolizing fatty acids released by neurons during excitotoxicity[42]. We showed that blocking FAS effectively depleted tissue state B signature from treated GBM slices. This may be explained by either a change in the composition of the GBM slices, given that FAS inhibition may lead to glioma cell death[32,41], a change of gene expression of the cells that reside in the slices, or both. The latter is likely the case, given that GBM slices treated with FAS inhibitor showed a positive enrichment for the gene signature of astrocytes treated with FAS inhibitor, and negative enrichment for tissue state B. These finding are clinically relevant, given FASN is a promising target against

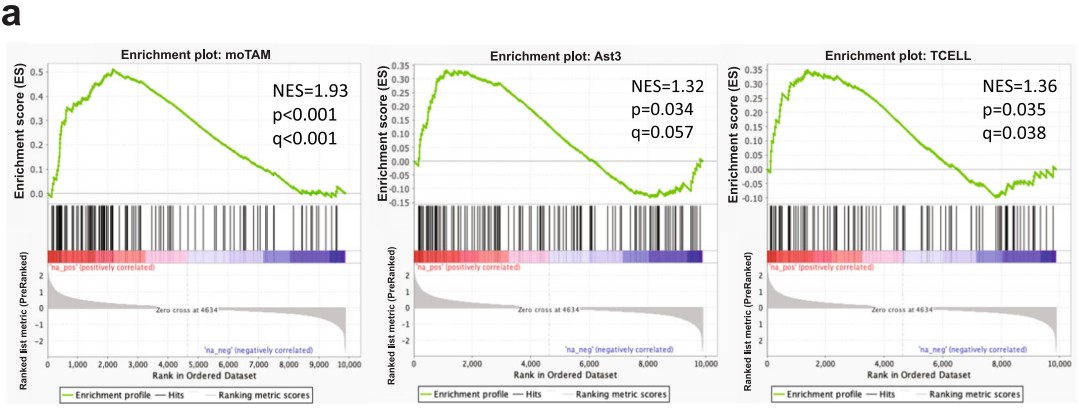

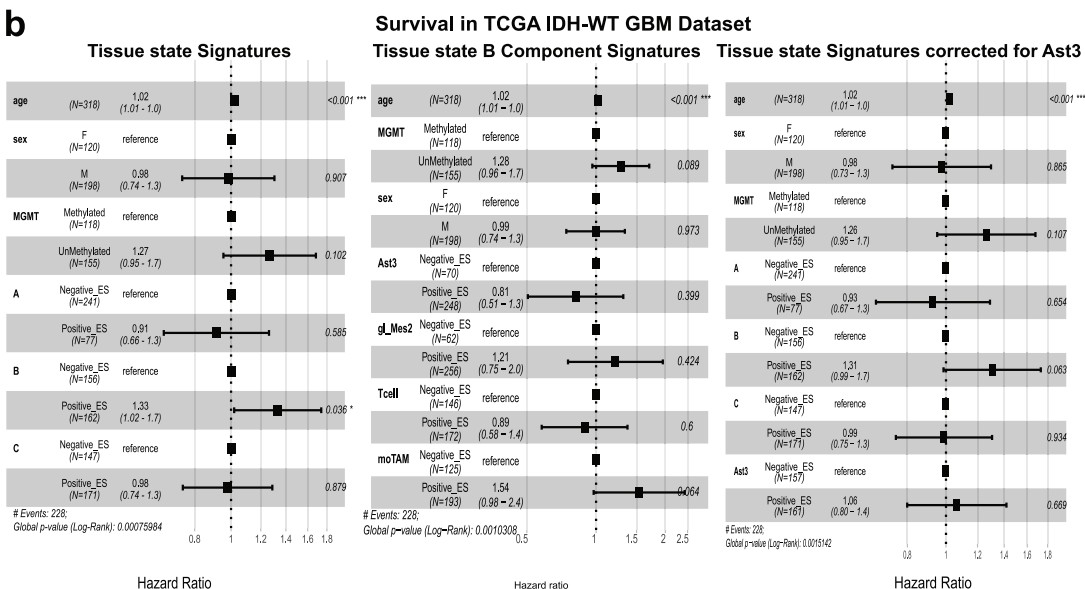

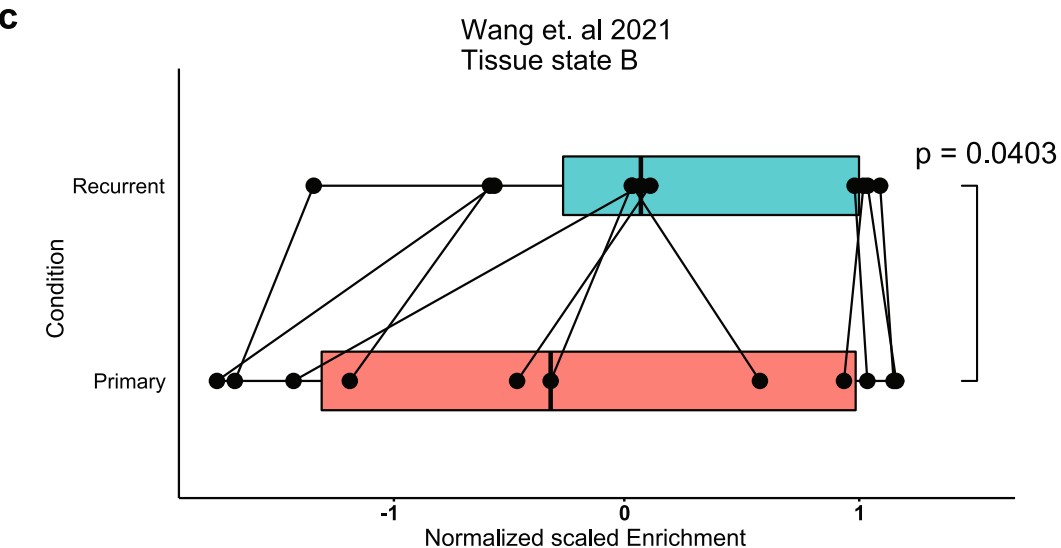

glioblastoma[43], and highlight how the tissue-state approach can provide insights into the effects of targeted therapies on the GBM microenvironment.

This study showed that the concept of tissue-states based on cellular cohabitation patterns generates testable hypotheses that inform our understanding of GBM biology. We found that tissue state B, which is enriched in reactive astrocytes (Ast3), monocyte-like tumor-associated myeloid cells, T-cells, and mesenchymal/astrocyte-like GBM cells (gl_Mes2) is associated with worse prognosis in GBM. Tissue state B is also characterized by specific metabolic signatures, like fatty acid metabolism, which we targeted ex vivo and showed it depleted the tissue state B signature. Future studies are needed to further evaluate the potential of targeting fatty acid synthesis to block GBM progression.

**Fig. 7 | Enrichment for tissue state B is independently associated with worse survival. a** Pre-ranked Gene Set Enrichment Analysis (GSEA) comparing tissue state B bulk RNAseq samples with tissue states A & C samples for 3 sub-lineages: Ast3, moTAM, and T-cells. Marker genes for each cell type were used as the gene set for each analysis. Normalized Enrichment Score (NES) is displayed, along with *p*-values and FDR-adjusted *q*-values. **b** Cox proportional hazard ratio of survival in the combined TCGA and CGGA IDH-WT GBM dataset given enrichment of each of the tissue state signatures (left), for the individual cell types that comprise Tissue State B (middle), and for each of the tissue states, regressing out enrichment of the Ast3 gene signature (right). Age, sex, and *MGMT* status are included as co-variates in the model. The *p*-values are shown on the left, bars indicate confidence intervals (also noted on the right). Enrichment of each geneset was categorized as negative or positive. **c** Boxplots of the tissue state B normalized enrichment scores in the Wang et al. (2021) paired primary and recurrent GBM dataset. Each box indicates the 25th, 50th, and 75th percentile enrichment scores per condition and paired samples are denoted by connected points. The whiskers indicate the minimum and maximum values. Significance was assessed using a one-tailed paired *t*-test, *n* = 11 per group. The *p*-value is indicated. Source Data are provided as a Source data file.

## Methods

All study protocols were approved by Columbia University Irving Medical Center Institutional Review Board.

### Human subjects and glioma tissue

Frozen primary untreated GBM tissue was acquired from the Bartoli brain tumor bank at Columbia University Medical Center. All diagnoses were rendered by specialized neuropathologists. Study protocols were approved by Columbia University Medical Center Institutional Review Board. All clinical samples were de-identified prior to analysis. Analyses were carried out in alignment with the principles outlined in the WMA Declaration of Helsinki and the Department of Health and Human services Belmont Report. Informed written consent was provided by all patients. The demographics of the cases used are provided in Supplementary Dataset 1. None of the participants were compensated for participating in this study.

### Extraction of nuclei and snRNAseq procedure

Nuclei were isolated from frozen surgical resection specimen in accordance with Al-Dalahmah et al.[23]. Briefly, the frozen tissue samples were dissected from fresh frozen tissue or frozen OCT-embedded tissue blocks to yield tissue measuring in general from $5 \times 2 \times 1$ mm to $10 \times 6 \times 3$ mm. The tissue was homogenized using a dounce homogenizer in ice-cold 30% sucrose 0.1% Triton-X 100 based homogenization buffer. 10–15 strokes of the loose dounce pestle were followed by 10–15 strokes of the tight dounce pestle on ice. Mixing using a P1000 pipette followed before filtration through a BD Falcon 40um filters. Filtration was repeated after a 10-min spin at $1000 \times g$ at 4 °C. A cleanup step followed using a density gradient step in accordance with[44]. The nuclear pellet was suspended in 1% BSA in PBS resuspension buffer containing RNAse inhibitors. A final filtration step using $20\,\mu m$ Flowmi™ filters followed before dilution to 700–1200 nuclei per μl in resuspension buffer. The nuclear suspensions were processed by the Chromium Controller (10x Genomics) using single-Cell 3′ Reagent Kit v2 or v3 (Chromium Single Cell 3′ Library & Gel Bead Kit v2, catalog number: 120237; Chromium Single Cell A Chip Kit, 48 runs, catalog number: 120236; 10x Genomics).

### Sequencing and raw data analysis

Sequencing of the resultant libraries was done on Illumina NOVAseq 6000 platformV4 150 bp paired end reads. Alignment was done using the CellRanger pipeline (10x Genomics) to GRCh38.p12 (refdata-cell-ranger-GRCh38-1.2.0 file provided by 10x genomics). Count matrices were generated from BAM files using default parameters of the DropEst pipeline[45]. Filtering and QC was done using the scater package (3). Nuclei with percent exonic reads from all reads in the range of 25–75% were included. Nuclei with percent mitochondrial reads aligning to mitochondria genes of more than 19% were excluded. Genes were filtered by keeping features with >10 counts per row in at least in 31 cells. Further filtering of low-quality cells was done to include cells with at least 400 detected genes and 10,000 reads.

### Single-nuclei RNAseq analysis

**Sequencing and analysis of raw data.** Sequencing of the resultant libraries was done on Illumina NOVAseq 6000 platformV4 150 bp paired end reads. We used 10X chromium v2 chemistry for samples PO1 and PO2, and v3 chemistry for samples PA1, PA2, and P3. Read alignment was done using the CellRanger pipeline (v3.1–10X genomics) to reference GRCh38.p12 (refdata-cellranger-GRCh38-1.2.0 file provided by 10x genomics). Count matrices were generated from BAM files using default parameters of the DropEst pipeline[45].

**Data-cleanup.** Filtering and QC was done using the scater package[46,47]. Nuclei with percent exonic reads from all reads in the range of 25–73% were included. Nuclei with percent mitochondrial reads aligning to mitochondria genes of more than 15% were excluded. Genes were filtered by keeping features with >10 counts per row in at least in 31 cells. The count matrix of each sample was normalized by first running the quickcluster function, then estimating size-factors by calling scran::computeSumFactors() function with default options and clusters set to clusters identified by calling quickcluster function. scater::normalize() function was then used to generated normalized counts. Doublet identification was done using scran::doubletCells function with default options, and cells with doublet score of NMADs >3 were excluded in accordance with previous publications[23].

**Combining multiple datasets from different sequencing batches.** To control sequencing and technical batches, we utilized canonical correlation analysis in Seurat[48] accounting for batch and mitochondrial read percentage for CNVneg nuclei. For CNVpos nuclei, we accounted for case and mitochondrial read percentage.

**Pre-Clustering and clustering of nuclei.** Pre-clustering of nuclei was done in Seurat using the shared nearest neighbor smart local moving algorithm. PCA reduction was used as the reduction in the Find-Neighbors() step. Pre-cluster identity determination was done using geneset enrichment analysis of lineage markers[23] and by inspecting cluster markers generated by scran::findmarkers(direction = "up") function. Microglia +/− oligodendrocytes were used as negative control cell for InferCNV pipeline (below). Once CNVneg cells were verified, cells from all cases were aligned using Seurat and clustered. Clusters with mixed identities based on enrichment of multiple lineage genes were sub-clustered iteratively until all "pre-clusters" showed pure identities. Only then do we combine the pre-clusters of the same lineage into lineages (astrocytes, neurons, oligodendrocytes, myeloid, endothelial). For subclustering of astrocytes and myeloid cells, we analyzed the nuclei in isolation of other lineages, and re-aligned them in Seurat, and reduced the dimensions before subclustering. For CNVpos nuclei, unbiased clusters were combined into glioma states/lineages based on similarity in marker expression and enrichment for known gene sets described in Supplementary Figs. 1d, 3d.

**Count normalization.** Raw counts were normalized in Seurat using the sctransform function SCT() function with default settings and controlling for percent mitochondrial gene expression[49]. We also corrected for the donor and sex.

**Copy-number variation analysis of snRNAseq.** To detect putative neoplastic tumor cells, we used combination of marker expression and large scale copy-number variation inference as per the InferCNV R

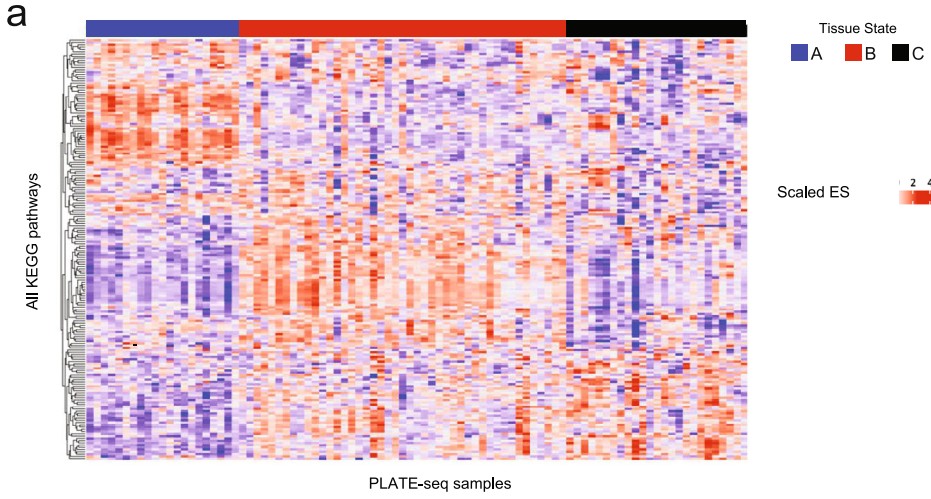

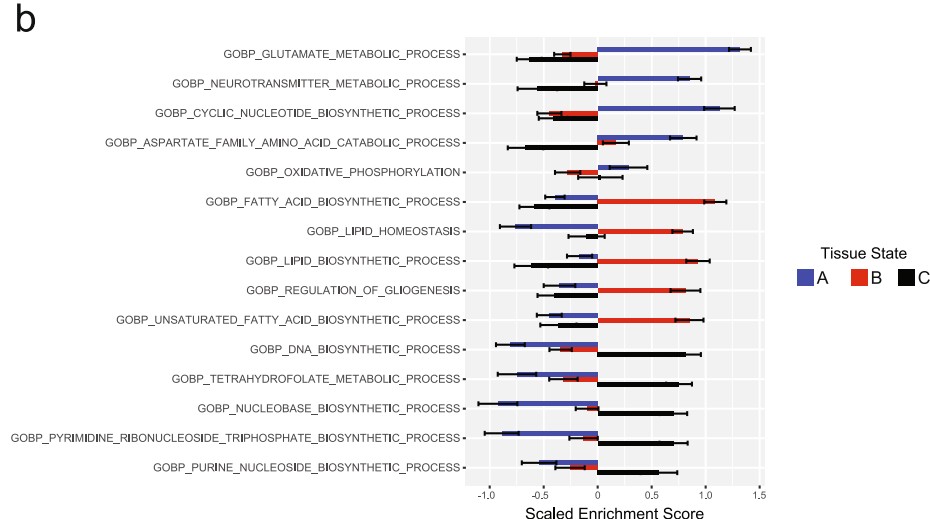

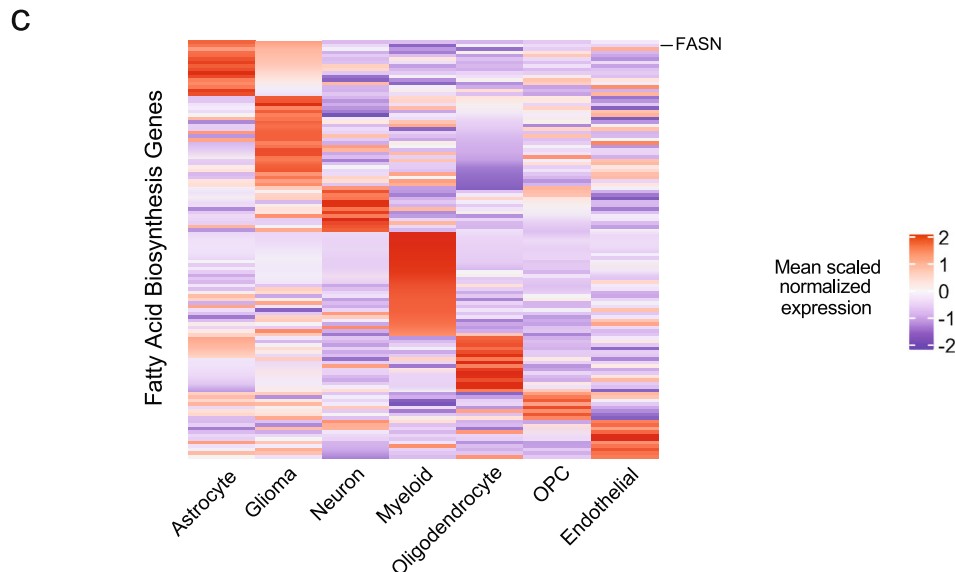

**Fig. 8 | Metabolic pathways drive targetable tissue state signatures. a** Heatmap displaying scaled enrichment scores for all KEGG pathways across all PLATE-seq samples. The heatmap is grouped by tissue state (cluster A, B, C), annotated by the horizontal bar at the top. Hierarchical clustering was performed on the rows (pathways), demonstrating cluster-specific metabolic programs. **b** Bar plot displaying scaled ssGSEA scores for select KEGG metabolic programs from panel **a**. Bar plots represent mean scaled ssGSEA score ± standard error for each of the three clusters for a given pathway. **c** Representative example showing a heatmap displaying mean lineage-specific scaled normalized expression of genes in the GO: Biological Process−Fatty Acid Biosynthesis gene set−which was most enriched in tissue state B. Note the expression of *FASN* is highest in astrocytes and glioma cells. Source Data are provided as a Source data file.

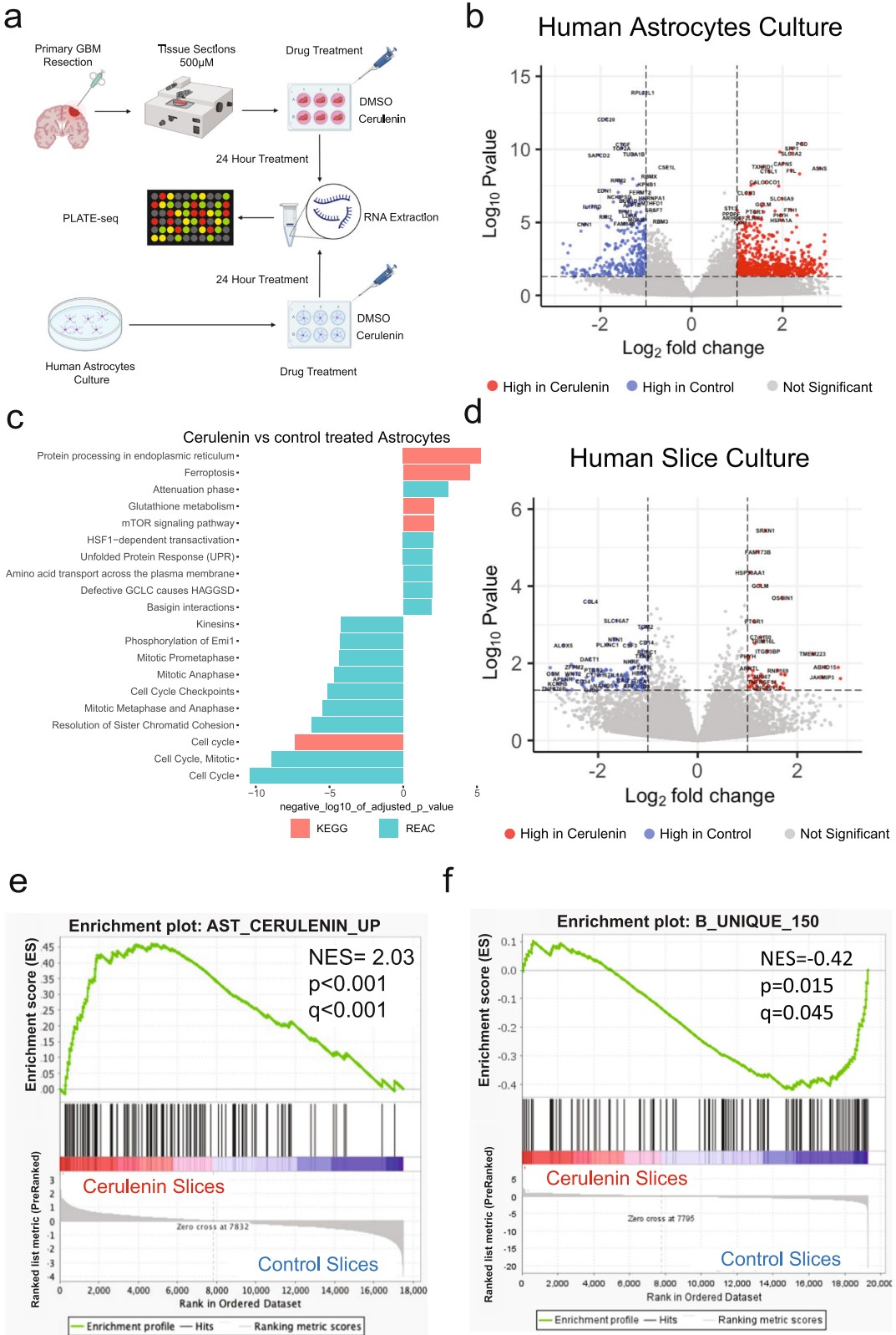

**Fig. 9 | Fatty acid synthase inhibition depletes tissue state B signature in GBM slice cultures. a** Scheme of in vitro and ex vivo fatty acid synthase perturbation studies. This panel was created with BioRender.com. **b** Volcano plot showing the log2 fold-change (x-axis) and log10 p-value (y-axis, two-tailed t-test) of differentially expressed genes in astrocytes treated with Cerulenin (5 mg/ml) vs. control. **c** KEGG and Reactome pathway enrichment analysis with the terms indicated on the y-axis, and the log10 p-value on the x-axis. The sign of the log10 p-value indicates the

direction of change (i.e., negative = reduced in Cerulenin treatment). **d** Volcano plot showing the log2 fold-change (x-axis) and log10 p-value (y-axis, two-tailed Wald-test) of differentially expressed genes in GBM slice cultures treated with Cerulenin (5 mg/ml) vs. control. **e**, **f** GSEA plots of pre-ranked enrichment of the genes increased in astrocytes treated with Cerulenin (**e**) and the top 150 genes unique to tissue state B signature (**f**). The normalized enrichment scores (NES), p-value (p), and adjusted p-value (q) are indicated.

package[50]. As a control population, we used microglia and oligodendrocytes from case PO2_1. Iteratively, CNVneg clusters including Oligodendrocytes and Neurons were identified and added as control cells. Different gene window sizes were tested (50, 100, 200) and yield similar results. We then applied an orthogonal approach to label putative neoplastic cells based on previous approaches described in refs. 2,51. Briefly, Log2 + 1 counts were averaged across chromosomes for each nucleus. A principal component analysis (PCA) was performed on autosomal chromosomes in factominer R package[28]. Chromosomes with high correlation with PC2 were the same as the ones shown in the detected by inferCNV() with the exception is PO1, where no neoplastic cells were detected by InferCNV or CONICS. A malignancy score was calculated by dividing the log2 gained chromosome counts over the sum of those that are lost (selection was limited to three chromosomes or less). The scores were then z-neoplastic per sample. To identify putative neoplastic nuclei in this method, we next performed $k$-means clustering of the scaled malignancy scores in R using the $k$-means function and centers argument set to 2. PO1 does not show bimodal malignancy score distribution and the results of $k$-means clustering were not considered. For case PA3, only a minority of nuclei had malignancy scores >2 standard deviations above the mean. Therefore, these cells were identified using outlier detection in a normal distribution as done in the getOutliers (method = "I", rho = c(0.1,3)) $iRight) in R. getOutliers is part of extremevalues R package https://github.com/markvanderloo/extremevalues. Only the consensus nuclei that were identified as CNV positive in both approaches were considered for analysis. Less than 7.0% of the nuclei were called alternately by the two methods and were excluded from the analysis. Identification of CNVpos nuclei in recurrent glioma samples was conducted through a combination of InferCNV and identification of clusters with high expression of tumor markers *SOX2* and *PTPRZ1*.

### Survival analysis
Survival analysis was performed using the survfit() function in the survival package in R[52,53], using the binarized enrichment of each of the gene sets as the covariate in the formula. For cox proportional hazards, the function coxph() was used in $r$, and the covariates are indicated in the main figures.

### Correlation analysis
Correlation analysis of glioma proportions was done using Pearson correlation (function cor() or psych::corr.test in R). Correlation heatmaps were generated using the corrplot package in R.

### Identification of glioma state and lineage top gene markers
The lineage-specific genes were determined using scater::findmarkers(…, direction = "up") function on the top-level lineages (Neurons, astrocytes, microglia, undetermined, oligodendrocytes, OPC, and endothelial cells). The glioma-state-specific genes were determined using scater::findmarkers (…, direction = "up") function on the neoplastic glioma states only. To select specific lineage/glioma-state markers, we further filtered the top markers generated above by selecting the genes with positive log-fold-change values in 90% or more of the cluster-to-cluster comparisons. The top 150 genes were selected and are provided in Supplementary Datasets 2–4 for primary glioma, recurrent glioma, and non-neoplastic lineages, respectively.

### Principal component analysis
PCA analysis was done in factominer R package[28]. A matrix of snRNA-seq sample by cell type/cluster was used as input (Supplementary Dataset 1). The proportion of each snRNAseq sample with respect to 11 non-neoplastic cell types (Ast1, Ast2, Ast3, Endothelial, mgTAM, moTAM, Myel1, Neuron, Oligodendrocyte, OPC, prTAM, Tcell) and the summed proportion of all neoplastic cell types (CNVPos) were used to form the initial PCA axes. The proportions of CNVPos cells that were

assigned to each of the glioma states (gl_Mes1, glMes2, gl_Pro1, gl_Pro2, gl_PN1, gl_PN2) were added as supplementary quantitative variables.

PCA coordinates of the bulk RNAseq validation dataset were generated using Factominer, by using the normalized expression of the genes filtered by rowsum > 1000.

### Acquisition of tissue and preparation of acute slice cultures
Primary GBM tissue from two separate surgeries, TB 6571 (3 blocks of tissue) and TB 6579 (2 blocks of tissue) (see supplemental table for related clinical information), performed at Columbia University Medical Center/New York Presbyterian Hospital were retrieved fresh from the operating room in a sterile specimen cup and transported back to the laboratory on ice. Primary GBM acute slice cultures were prepared[54]. 500 μm thick slices were treated with either DMSO or 5 μg/ml Cerulenin for 18 h prior to preservation and RNA extraction.

### Bulk RNAseq using PLATE-Seq and from public datasets (TCGA and CGGA)
RNA extraction was done using the RNeasy Mini Kit (Qiagen cat# 74106). RNAseq was performed on MRI-localized biopsies ($n = 91$) using PLATE-seq[55] as described below and in accordance with ref. 55. Specifically, 75 bp paired end sequencing was performed on Illumina NextSeq platform. The first read was used to extract the barcode, error-correct it, and map it to a known barcode sequence space. The second read was aligned using STAR[56] to the human genome (hg19, annotation: UCSC known genes). STAR-generated read counts were used to generate FPKM values, which were used in GSEA analysis. The count matrix for the TCGA GBM dataset was downloaded using the GDCquery tool in R. The Chinese Glioma Genome Atlas (CGGA) RNA-seq datasets[57,58] was downloaded from (http://www.cgga.org.cn/download.jsp). The counts were normalized using the vst() function in deseq2 R package[59]. IDH-WT only samples were kept from both datasets (TCGA: 139 samples, CGGA: 179 samples) and used for downstream analysis.

For acute slice-culture PLATE-seq analysis, slices were then transferred to OCT and frozen into blocks. Tissue from each slice was isolated for RNA extraction by the Columbia Molecular Pathology Core using QiaSymphony extraction method. Total RNA was quantified using Nanodrop measurements, and 150 ng of RNA from each slice/condition was loaded into a well of a 96 well PLATE. Pooled library amplification for transcriptome expression (PLATE-Seq)[55] was then performed on the 96 well plate and the resulting data analysis was done as described above. U87 and Astrocyte co-culture PLATE-Seq was conducted the same way as described above.

### Differential gene expression analysis
For comparing astrocytes to glioma, EdgeR glmQLFTest was used and the top 3000 differentially expressed genes with an FDR cutoff of 25%[60] were extracted. Only datapoints with adjusted $p$-values less than 0.05 were used in downstream analysis. For PLATE-seq data differential gene expression analysis between treatment and control was performed adjusting for tissue block and patient using the Deseq2 pipeline[59]. For astrocyte cultures, differential gene expression analysis between treatment and control was performed adjusting for astrocyte passage and cell culture batch using the Deseq2 pipeline.

### Geneset enrichment analysis and gene ontology analysis
The average normalized counts per gene per cluster was calculated. The resultant cluster-wise count matrix was used as input to the GSVA pipeline[61]. Gene sets used for various tests are provided in the supplementary material (Supplementary Datasets 2–4). The options used for performing the GSVA pipeline are as follows: method= ssgsea, kcdf = "Gaussian", mx.diff=TRUE. Heatmaps were generated using the heatmap.2 in R function from the package gplots (R Package) and

scores *z*-scaled were indicated. Ontology enrichment analysis in gProfiler with default settings[62]. For GSEA of the combined TCGA and CGGA dataset and the validation bulk RNAseq dataset (Fig. 3C) and Fig. 3I, the enrichment was performed using method = "gsva" option on normalized counts, which normalizes the enrichment scores for each gene-set per sample. GSEA analysis in Figs. 7 and 9 was conducted using pre-ranked GSEA and was performed according to Subramanian et al. with 1000 permutations[63]. Log-2-fold-change between cluster B and the remaining clusters was used to rank the genes for the analysis, and marker genes from each sub-lineage was used for the gene sets. For GSEA in Fig. 9e, f, pre-ranked GSEA (based on log2fold-change) was performed using tissue cluster gene sets and the genes significantly upregulated after astrocyte treatment with cerulenin or the top 150 genes unique to tissue state B signature.

## Generating the tissue state signatures

Pseudo-bulk samples from the snRNAseq dataset were created by summing and rounding the normalized counts per sample. Differential gene expression analysis using the DESseq2 pipeline was conducted between the clusters, controlling for sequencing batch (Supplementary Dataset 3). The genes significantly differentially increased in cluster C vs. A and C vs. B constituted tissue state C signature. The genes significantly differentially increased in cluster B vs. A and B vs. C constituted tissue state B signature. The genes significantly differentially increased in cluster A vs. C and A vs. B constituted tissue state A signature. Clustering samples of the validation bulk RNAseq dataset into the three tissue states was performed by first retrieving the geneset enrichment scores of the genes unique to each tissue state in each sample using the gsva algorithm with method = "gsva". Next, we performed hierarchical clustering on the Euclidian distance matrix calculated from the gsea scores, with method = "Ward.D2" in hclust, and cutree function with $k = 3$—all in R.

## Spatial transcriptomics

Spatial transcriptomics was conducted using 10X™ Visium Spatial Gene Expression Slide & Reagent Kit, 16 rxns (PN-1000184), according to the protocol detailed in document CG000239_RevD available in 10x demonstrated protocols. 10 micron-thick tissue sections were mounted on the ST slides and stained for nuclei—DAPI among other antigens using a rapid immunofluorescence protocol described in document CG000312_RevB available in 10X demonstrated protocols. Imaging of whole slides was done at 20X magnification on a Leica Aperio Versa scanner or a Leica DMI6 thunder tissue imager. After imaging, the slides were de-cover-slipped and the tissue permeabilized for 11 min (which was empirically determined to yield best results based on the Visium Spatial Tissue Optimization Slide & Reagent Kit PN-1000193 as detailed in the protocol provided in document CG000238_RevD available in 10X demonstrated protocols). The remaining steps were conducted according to the manufacturer's protocol. The libraries were sequenced on multiple Illumina Nextseq 550 (paired end dual-indexed sequencing) flowcells to achieve the recommended number reads per ST spot. The spatial transcriptomic (ST) samples were prepared using 10X genomics Cell Ranger (version 6.1.2) and Space Ranger (version 1.2.1) software. Raw tiff images of the tissue were labeled with Cell Ranger which generated a json file for Space Ranger to use during alignment. Labeled spots from Cell Ranger were inputted into the loupe-alignment argument in Space Ranger along with its respective tiff image file, FASTQ reads, and slide numbers. The reference genome used for alignment was built using the Space Ranger function spaceranger mkgtf with GRCh38 as the assembly and Ensemble 91 for the transcript annotations. All other parameters to generating the counts data for ST were set to its default setting. The number of counts per spot per ST sample is shown in Supplementary Fig. 10c. The plots of ST experiments shown in Fig. 5a and Supplementary Fig. 11 were generated using SPATA2[64].

## Deconvolution and spatial cross-correlation analysis

Deconvolution using *RCTD* was used to determine the proportion of each cell type at each spot in each of the 9 ST experiments[29]. *RCTD* was run in "full" mode and used the complete annotated set of single nuclei ($n = 43,505$) as a reference. The differential gene expression threshold in the "createRCTD" step was set to 1.25 logFC; other parameters were set to their default values. Bulk samples were deconvolved by supplying null coordinates Deconvolution performance was quantified using immunohistochemical staining of the ST samples for DAPI (Supplementary Dataset 1). Using the package BayesSpace, spots in each experiment were clustered according to their transcriptional profiles and cartesian coordinates. The number of BayesSpace clusters in each sample was determined using the SC.MEB package with criterion set to "MBIC"[65]. The mean proportion of each cell type was calculated within each BayesSpace cluster (total clusters $n = 33$). Using QuPath cell detection, cells were segmented and then assigned to each BayesSpace cluster using the st_within function (R package *sf*). The number of DAPI-positive nuclei within a BayesSpace cluster was computed and divided by the area of the BayesSpace cluster to obtain the nuclear density. The density was then correlated with the mean proportion of each cell type and CNVpos (the sum of all neoplastic cell types) using the cor.test function.

Cohabitation patterns between cell types in the ST data were quantified using spatial cross-correlation as implemented in the R package *MERINGUE* and evaluated at neighborhood size = 900 μM[66]. To determine the spatial adjacency matrix for the first order calculation (a spot's immediate neighbors), the Cartesian coordinates of each Visium spot were input into the "getSpatialNeighbors" function. As Visium spots are arranged in a hexagonal lattice, the parameter "filterDist" was initially chosen such that no spot had greater than six contiguous neighbors. Spatial adjacency matrices were created using the *igraph* package by generating a graph from the first order spatial adjacency matrix using the "graph_from_adjacency_matrix" function and then inputting this graph into the "connect" function[67].

The deconvolved proportions of each cell type output by *RCTD* (summed to unity on a spot-by-spot basis) and the first order spatial adjacency matrix were used as inputs for the "spatialCrossCorMatrix" function in *MERINGUE* to determine the average pairwise spatial cross-correlations between each spot and its neighbors between each of the 18 cell types (171 combinations total) in each experiment, The pairwise comparisons were normalized with respect to the number of spots in each sample before being averaged across samples and plotted using the *ComplexHeatmap* package[68]. The significance of the cross-correlations was determined using the "spatialCrossCorTest" function with 100 permutations: the spatial cross-correlation calculation was repeated 100 times using neighborhoods consisting of randomly selected spots from that sample. The proportion of random permutations that yielded spatial cross-correlations at least as high as those obtained from the actual data was taken as the *p*-value for that relationship. For each sample, adjusted *p*-values for each cross-correlation relationship were determined using Benjamini–Hochberg correction. Significance across samples was computed using the Fisher method for combining *p*-values across independent experiments (*poolr* package, R, unweighted). Relationships with a combined adjusted *p*-value less than 0.05 were considered significant. Dendrograms for each of the resultant heatmaps were determined using ward.D clustering of the Euclidean distance between the spatial cross-correlation values for each cell type.

## Immunohistochemistry, histology, and in situ hybridization

Standard chromogenic Immunohistochemistry was done as described below and in accordance with ref. 23. Briefly, paraffin-embedded formalin-fixed tissue sections or fresh frozen sections briefly fixed in 4% PFA, for 10 min (40 °C) in 4% PFA in PBS. Paraffin sections after deparaffinization were treated with antigen unmasking solution

according manufacture recommendations (Vector Laboratories, Burlingame, CA). The following antibodies and dilutions were used SOX2 (1:200, Mouse monoclonal, Abcam, Ab218520), KI67 (1:500, rat monoclonal polyclonal, Thermo Scientific, 14-5698-80), CD68 (1:200, mouse monoclonal, Abcam cat# ab955), NeuN (1:1000, mouse monoclonal, Millipore, MAB377). For fluorescent IHC, secondary antibody conjugated to fluorophores: anti-mouse Alexa Fluor 488 and 594 and anti-rabbit Alexa Fluor 488 and 594; goat or donkey (1:300, ThermoFisher Scientific, Eugene, OR) were applied for 1 h at room temperature. In situ hybridization was done using RNAscope™ multiplex fluorescent v2 (ACDbio cat# 323100) per the manufacturer's protocol in 5-micron paraffin-embedded, formalin-fixed tissue sections. We used predesigned probes for PTPRZ1, CLU, TOP2A, NOVA1, MEG3, and SOX2 from ACDbio; cat# 584781, 584771, 470321, 400871, 584801, and 400871, respectively. Fluorescent images were taken on a Zeiss 810 Axio confocal microscope at 40X. Brightfield fluorescent images were taken on an Aperio LSM™ slide scanner at 20x and 40x.

## Quantification of ISH

For quantification of in situ hybridization images we used the positive cell detection function in Qupath v0.2.3[69]. We only quantified signal contained in DAPI-positive nuclei. First, DAPI-positive nuclei were detected using the cell detection tool. Next, subcellular detection function was employed to segment puncta per each of the three probe channels. A random tree classifier was used to classify nuclei to be positive or negative in QuPath under default settings, with a minimum of two puncta per channel to classify a nucleus as positive for the probe. Infiltrated cortex and cellular tumor core were annotated by a neuropathologist.

## Cell culture and co-culture

Human Astrocytes (ScienCell cat #1800) were cultured in Astrocyte culture medium (ScienCell cat# 1801), 2% fetal bovine serum (ScienCell cat 0010), 1% astrocyte growth supplement (ScienCell cat# 1852) and 1% penicillin/streptomycin (ScienCell cat # 0503). The cells were maintained as adherent cultures on poly-L-Lysine coated tissue culture plates. The cells were passaged at 70–90% confluence and treated at passage numbers 5–7. DMSO or Cerulunin Sigma cat#C2389 at 5 μg/ml was used to treat the cells for 18 h as indicated.

Human astrocytes used co-culture were first transduced with lentiviruses carrying *GFP* (LentiORF control particles of pLenti-C-mGFP-P2A-Puro Origene™ cat# PS100093V), *CLU* (Lenti ORF particles, *CLU* (mGFP-tagged)-Human Clusterin (CLU), transcript variant 1 Origene™ cat# RC211875L4V), or *LGALS3* (Lenti ORF particles, LGALS3 (mGFP-tagged)—Human lectin, galactoside-binding, soluble, 3 (LGALS3), transcript variant 1, Origene™ cat# RC208785L4V). Transduction was performed by inoculating astrocytes seeded at $10^4$ cells per well with 10 μl of virus at $1 \times 10^7$ TU/ml (5 MOI), in the presence of 10 μg/ml polybrene, followed by 1-week selection in a 0.5 μg/ml puromycin containing selection medium. Transduction efficiency was confirmed by observing fluorescence on microscopy and FACS analysis. Co-culture with U87-MG (obtained from ATCC—maintained in DMEM + 10%FBS) ensued for 24 h—with both Astrocytes and U87-MG cells seeded at $2*10^5$ cells/well—6-well plate). The cultures were trypsinized 24 h after seeding and subjected to FACSorting (influx cell sorter—Beckman Coulter, Jersey City, NJ), into GFP+ (astrocyte) and GFP− (U87MG) fractions, from which RNA was extracted as described above.

## Real-time quantitative PCR

Total RNA was extracted from brain specimens using RNAeasy minikit (Qiagene©). RNA concentration and purity were determined using NanoDrop (Thermo Scientific™, MA). RNA was converted to cDNA using High-capacity RNA-to-cDNA kit (Thermo Fisher Scientific, Applied Biosystems™, MA). The following Taqman assays were used

(*CHI3L1*-Hs01072228_m1, *CD44*-Hs01075864_m1, *LGALS3*-Hs00173587_m1, *GAPDH*-Hs02786624_g1, *MIB1*-Hs01075903_m1, HES5-Hs01387463_g1, CLU-Hs00156548_m1, EZH2-Hs00544830_m1, *SOX2*-Hs04234836_s1, *NES*-Hs04187831_g1, *HES1*-Hs00172878_m1). The reaction volumes were 15 μl per reaction. TaqMan™ Multiplex Master Mix (Thermo Fisher Scientific cat# 4461881) was used. All reactions included 5 ng of cDNA. Thermal cycling parameters were conducted per manufacturer's standard recommendations. The qPCR plates were read on a QuantStudio™ 5 Real-time PCR system (Thermo Fisher Scientific, Applied Biosystems™, MA). The reactions were done in triplicates. Relative gene expression was calculated using the delta delta Ct method with GAPDH as a reference gene.

## Statistical testing

Statistical comparisons were done using one-way ANOVA (or Kruskal–Wallis test) and Tuckey post-hoc comparison in R. Statistical testing for RNAseq application is reported in the main text or respective methods section. Differential abundance analysis was done employing a moderated regression model in ANCOMBC with default parameters, assigned Condition (primary vs. post-treatment recurrence) and CNVpos proportions in the design formula, and in accordance with the authors guidance[27]. One-tailed paired *t*-tests were done to compare the core and margin percentages of the same case (Supplementary Fig. 5). A one sample *t*-test was conducted to determine if the percentage of TOP2A+ that were CLU+ was less than 50%. One or two-tailed *t*-tests were used in Supplementary Fig. 13 as indicated in the legend.

## Reporting summary

Further information on research design is available in the Nature Portfolio Reporting Summary linked to this article.

## Data availability

Data for spatial transcriptomic can be queried using an interactive web app: https://vmenon.shinyapps.io/gbm_expression/. The data generated and/or analyzed for this study are included in this article, its supplementary information files, and on GEO GSE228500. The data on GEO has all raw data for snRNAseq and ST samples as indicated in supplementary datasets 1, and bulk RNA-seq samples as indicated in supplementary datasets 6, 8 and referring to Fig. 6. The processed single nuclei and spatial transcriptomics data are available on [https://github.com/adithyakan/reconvolving_gbm]. The DEG and enrichment analysis data generated in this study are provided in the Supplementary Information/Source Data files. The Cancer Genome Atlas (TCGA) data was accessed using the TCGAbiolinks package (version 2.18.0). Chinese Glioma Genome Atlas (CGGA) RNAseq datasets can be downloaded from [http://www.cgga.org.cn/download.jsp]. Datasets from Gill et al. (2014) can be accessed through GEO: GSE59612).

## Code availability

Custom code and processed data objects are available on github: [https://github.com/adithyakan/reconvolving_gbm].

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

## Acknowledgements

The results published here are in part based upon data generated by the TCGA Research Network: https://www.cancer.gov/tcga. This research was funded by NIH/NINDS R01NS103473 (P.C., P.A.S., J.N.B.), The William Rhodes and Louise Tilzer-Rhodes Center for Glioblastoma at New York-Presbyterian Hospital (O.A., V.M., P.C.), Herbert Irving Comprehensive Cancer Center pilot research grant (O.A.), NIA ADRC REC program Grant Number P30AG066462 (O.A.), and the NIH/NCI Cancer Center Support Grant P30CA013696 (O.A., A.B.L., J.N.B., P.A.S., P.C.). This research was supported by the Genomics and High Throughput Screening Shared Resource and the Digital Computational Pathology Laboratory in the Department of Pathology and Cell Biology at Columbia University Irving Medical Center. We thank Dr. Claudia Deoge for help with RNAscope. We thank the immunohistochemistry core in the Department of Pathology and Cell Biology at Columbia University Irving Medical Center for help with GFAP-SOX2 immunostains. We thank Al-Fardthakh© for support with aspects of bulk RNAseq data analysis. This publication was supported by the National Center for Advancing Translational Sciences, National Institutes of Health, through Grant Number UL1TR001873. The content is solely the responsibility of the authors and does not necessarily represent the official views of the NIH.

## Author contributions

O.A., J.N.B., P.A.S., V.M., and P.C. designed the study; O.A., M.A., A.K., D.B., A.S., A.M., F.K., J.F.T., A.R.G., A.D., M.A.B., J.F., T.S., E.B., N.H., J.N.B., and P.C. conducted the experiments, including tissue procurement, tissue processing, and tissue analysis. J.L. prepared sequencing libraries and performed the experiments. H.L. performed the cell culture experiments. J.N.B., A.B.L., G.M.M., B.J.A.G., B.Y., and M.B.S. diagnosed and recruited the patients. Y.S., F.P., and A.K. perfomed the analysis on spatial transcriptomics datasets. O.A., M.A., A.K., P.A.S., V.M., and P.C. analyzed the data. O.A., M.A., A.K., V.M., and P.C. wrote the manuscript and all authors edited, read, and approved the final manuscript.

## Competing interests

P.A.S. receives patent royalties from Guardant Health. Columbia University has filed a patent application on the microwell single-cell RNA-seq technology used in this study, and P.A.S. is listed as a co-inventor. The patent number is WO/2016/191533. The patent title is "RNA printing and sequencing devices, methods, and systems". None of the other authors declare any competing interests.

## Additional information

[1]Department of Pathology and Cell Biology, Columbia University Vagelos College of Physicians and Surgeons (VP&S), New York, NY, USA. [2]Herbert Irving Comprehensive Cancer Center, Columbia University Irving Medical Center, New York, NY 10032, USA. [3]Department of Neurological Surgery, Columbia University Vagelos College of Physicians and Surgeons (VP&S), New York, NY 10032, USA. [4]Department of Neurology, Columbia University Vagelos College of Physicians and Surgeons (VP&S), New York, NY 10032, USA. [5]Department of Systems Biology, Columbia University Vagelos College of Physicians and Surgeons (VP&S), New York, NY 10032, USA. [6]These authors contributed equally: Osama Al-Dalahmah, Michael G. Argenziano, Adithya Kannan. ✉e-mail: oa2298@cumc.columbia.edu; vm2545@cumc.columbia.edu; pc561@cumc.columbia.edu

