## [Peer review file · Nature Communications]

Reviewers' comments:

Reviewer #1 (Remarks to the Author): Expert in glioblastoma genomics and intratumour heterogeneity

Re-convolving the compositional landscape of primary and recurrent 1 Glioblastoma Using Single nucleus RNA sequencing

Submission to Nature Communications

Al-Dalahmah et al. submit a report regarding the compositional landscape of primary and recurrent GBMs. The question of what cellular and molecular features define differences between primary and recurrent GBMs is under active research, with multiple groups and consortia such as the GLASS consortium active in this area. The question is an important one, since clinical therapeutic options at recurrence are limited, and a better understanding of how pathophysiology differs at recurrence would help to improve therapeutics.

The investigative goals of this study were only broadly defined - the two main goals of the study appeared to be

- 1) How tumour composition changes between primary and recurrent tumors
- 2) to use snRNA as a more comprehensive method of analyzing what types of cells play a role in tumour margins.

No clear hypothesis was defined a priori - for example it was not made explicit what type of cell or process was hypothesised for being responsible at recurrence, and although it was stated that snRNAseq provided more detailed cellular compositional view due to its preservation of astrocytes and neurons, a potentially valid point, there was no hypothesis as to why investigation of these cells would be relevant and overarching goal seemed to be to provide a description of cell compositions.

In terms of experimental approach, although snRNA seq does provide certain advantages - e.g. ability to analyze banked samples, it also has drawbacks, e.g. the loss of non-nuclear RNA information (which may or may not be relevant). Furthermore, although returning to frozen banks is a definitive advantage, a key weakness of this study is the low number of matched primary/recurrent pairs, which the authors acknowledge. Without primary-recurrent pair testing, and given the heterogeneity of the TME, comparing a series of distinct primaries with distinct recurrences loses some of its value. Furthermore, one of the explicit goals is to explore the nature and cellular composition of margin tissue. Yet information as to the spatial location of the samples is not made clear from the methods (were these marginal tissue samples?). Even in Supplementary Table 1, it is not clear what the locational provenance of the samples analyzed is. This weakness was partially remedied through spatial analysis and re-analysis of previously collected bulk seq data with spatial information, however given the unclear provenance of

the initial discovery dataset, any interpretation which transfer findings from the snRNA dataset must be treated with caution.

In terms of patient selection, lower grade gliomas were included, which consisted of two oligodendrogliomas, and an anaplastic astrocytoma; the rationale for including these in the analysis is again not clear as oligos represent quite a distinct pathological entity, and having IDH mutant primaries without recurrences again makes comparison and interpretation difficult.

The finding that recurrent tumours have a more mesenchymal state is interesting and corroborates what others such as Wang et al. have noted. Indeed the proneural to mesenchymal shift with its attending changes in the immune environment can probably be seen as a defining feature of recurrence.

Astrocytes - the analysis of tumour associated astrocytes was an interesting investigation, as noted previously these cell types are infrequently studied. 9 clusters of these cells were identified from 707 cells. Such a low number of cells means that the significance of the marker genes from each cluster is limited. The identification of cluster 6 as one which was associated with prognosis was made. The genes CD44, CHI3L1, LGALS3, CLU and APOE were identified as genes marking this cluster, however no attempt was made to understand how these genes relate to each other and why they would have prognostic relevance. Indeed, having some form of astrocytic gene signature correlate with poor outcome in astrocytomas is not surprising, but it would be useful to know whether this prognostic relationship has any basis in biological studies or tissue localisation. Importantly a more systematic comparison to normal astrocytes and the heterogeneity of the latter, would have been valuable.

The use of the re-convolution technique in the study seemed like a superfluous exercise to a good extent. As far as I can make out the process is the reconstruction of sample cellular composition by aggregating all the cellular states and splitting them using dimensionality reduction to find 3 predominant "tissue states" across samples. These tissue states describe a cell type grouping which can be found across samples and correspond to infiltration, reactive changes and proliferation. How does this differ from just looking at the sample cellular composition using immunohistochemistry? One advantage may be to better define the cellular composition characteristics and the cross-talk and interactions which occur within each of these tissue states. However this was not explored. Tissue states were correlated with prognosis, and again the finding that a mesenchymal signature is associated with worsened prognosis does not put us any further than we were before.

Enrichment of metabolic pathways was analyzed, and again although potentially interesting leads were highlighted, such as the glutathione pathway, none were explored even preliminarily.

Conclusion

The use of snRNA from frozen tissue and tissue banks is an important technical advance which allows study of longitudinal samples and interrogation of existing biobanks, and the authors should be commended on obtaining data of sufficient quality to perform detailed analyses. However the study overall suffers from a lack of focus - if the aim of the study was to study primary and recurrent differences then matched pairs would provide for a better experimental design and tissue substrate. If however the goal was to study margin tissue, then a more targeted approach in tissue collection, with specific reference to how and where tissue was sampled from would be better suited to answering this question. Furthermore methods such as re-convolving and assembling tissue states by clustering of cellular states does not give additional information beyond what is already seen and observed using standard immunohistochemistry. The observation that certain cell types tend to cluster together in tumours into infiltrating regions, reactive regions and cellular regions is not a novel finding, and although confirmed with spatial studies it does not get us any further in terms of understanding how these cell types interact to cause disease or recurrence. This study represents a somewhat superficial descriptive study of an important area, and the findings and the techniques employed do not advance our understanding in an appreciable way.

Reviewer #2 (Remarks to the Author): Expert in glioblastoma genomics and single-nucleus RNA-seq

Al-Dalahmah et al. present single-nucleus RNA-seq and other omics analyses of primary and recurrent glioma specimens. They describe cell types and states found in tumor cells and in the microenvironment and validate some of their observations in situ via spatial transcriptomics and IF. The manuscript is of interest due to the comparative analysis of cell composition between primary and recurrent specimens. There are several minor to moderate issues that decrease enthusiasm. I would ask the authors to consider the following points:

1. The sample size is somewhat low, if the authors can increase that it would improve their power.
2. The authors have validated their findings using 91 published primary and recurrent GBM bulk RNA-seq datasets. However, there are several hundred bulk GBM datasets at least which contain primary and recurrent specimens in the literature. I would ask the authors to increase their sample size. Some power analysis would increase confidence.
3. The authors have mostly used the bulk RNA-seq to confirm the microenvironment gene expression. I would ask the authors to also confirm some of their other findings in bulk RNA-seq data.
4. Please provide QC metrics on the snRNA-seq in the supplement, including the distribution of number of genes tagged and doublet rates. Along these lines, did the authors remove doublets? They should if they haven't.

5. The mesenchymal and proneural signatures don't seem to contain canonical markers. Please plot some more standard markers (CD44, CHI3L1 for mes, and OLIG2, ASCL1 for proneural for example). Moreover, it is not clear why the proliferative state is separate from these other states. Previous studies show that both the proneural and mesenchymal cell types proliferate. Can the authors compute a proliferation score say via Seurat and score their other cells to see which compartments are cycling.
6. The microenvironment signatures from astrocytes are interesting but no validation has been done that these factors are elaborated by neoplastic cells or astrocytes, or that they promote malignant progression or are therapeutically relevant beyond correlation. Any work done here increase the significance of the manuscript and any novel targets and demonstration of being able to modulate viability or survival by targeting these signals would increase the value of the manuscript.
7. Very little has been done to integrate the plate-seq data and snRNA-seq from the same patient. Moreover, the plate-seq data is a small sample size (n=1). As it stands this data does not really add much since it is underpowered and yet not integrated with the other data in the manuscript.
8. The fonts on most of the figure panels are way too small to read, especially the GO lists and gene lists on the heatmaps. It is not necessary to provide a long list of GO terms in a figure. This can be in a supplementary table if necessary. Either way, the figure quality needs to be greatly improved.
9. The survival analyses in Figure 4-5 are not controlled for age and sex, therefore should be viewed as preliminary. Also, it is not clear if they are controlled for IDH-mutation status which is critical.

Reviewer #3 (Remarks to the Author): Expert in glioblastoma tumour microenvironment and astrocytes

The presented manuscript titled: "Re-convolving the compositional landscape of primary and recurrent Glioblastoma Using Single nucleus RNA sequencing" reports a mapping of snRNA sequencing of a total of 21 samples. The major goal was to investigate the tumor microenvironment from primary and recurrent GBM. Especially large cells such as human astrocytes or neurons are normally underrepresented based on the nozzle size of the FACS or the 10X technology itself. The approaches and data sound promising at first, but the presentation and interpretation disappoint and make it difficult to see the connections. A clear story and thread are not evident in the manuscript and hopping between sections makes comprehension difficult.

Major concerns:

- The overall quality of the data is difficult to evaluate because of the lack of information on the average number and genes per cell and other quality characteristics.
- A key data issue arises from the lack of a horizontal data integration concept. It has been known for years that data integration of malignant samples can be challenging. This is true for both primary and secondary GBM samples. If a direct comparison of disease status is desired, sufficient integration of all snRNA-seq data must first be performed (e.g., Mutual Nearest Neighbor, CCA is not recommended in

this context). Then malignant and non-malignant cells can be better distinguished and the differences between non-malignant TME and disease status become clearer.

- Malignant cells identified by clusters and/or InferCNV, etc., present another challenge. If the individual patients cannot be easily integrated (which is the case for most datasets), other analysis approaches must be taken (see Neftel et al.), and uniform transcription programs should be sought.

- Here, the presentation of the data lacks information about the patients in each identified cluster. For UMAP, I would strongly suspect that the identified groups are primarily patient-specific (Figure 1).

- The yield of the experiments sounds very low: 15189 nuc from 8 samples when using the 10X system seems very little to me. How did this happen? were several patients loaded per lane? How many nuc. were loaded per lane? The expected number would be between 60tds and 80tsd nucs.

- In the first part, the authors identified transcriptional subtypes that largely overlap with known states. Direct comparison with the established classification (Neftel) is lacking. The authors should also consider using the Neftel states as a reference map, as the main goal of this work is not a novel single-cell classification.

- From the H&E and the CNV profiles, the margin samples seem to cover different areas from infiltrating areas to normal brain (PO1). How where the samples guided and planned are the points saved in the Neuronavigation system? It would be interesting to show the distance to the contrast enhancing region for each “margin” sample.

- The authors showed that similar states and state distributions were observed in primary and recurrent samples. Why not integrate both types of data and show the similarity?

- The concept of spatial heterogeneity and microenvironmental crosstalk sounds interesting but is not explored further. The addition of spatial transcriptomics with single cell integration of the snRNA-seq dataset (Cell2Location/Spotlight) will add significant value.

- The greatest advantage of the manuscript is the microenvironment data. It is a pity that these data are not presented more concisely and that the focus is on microenvironment changes between primary and relapse status. The crosstalk of myeloid and glial cells plays a crucial role and should be more clearly elaborated here. Extrapolation of signatures to the TCGA database is not very helpful and does not provide new insights.

- Direct crosstalk between glial and myeloid cells causing reactive transformation can be explored in the context of disease status. CellChat or other tools can be used to study the cellular interactions. Cellular evolution from a healthy cell to different activation states should be studied using RNA velocity or other

tools to reconstruct the evolution of cell lines. Unfortunately, the authors did not use most of the established tools that help in the in-depth study of the datasets.

- The last part aims to examine tissue states that may correlate with clinical parameters. The concept does not become clear to me and cannot be addressed with the data shown.

Reviewer 1 Comments

Comment: Al-Dalahmah et al. submit a report regarding the compositional landscape of primary and recurrent GBMs. The question of what cellular and molecular features define differences between primary and recurrent GBMs is under active research, with multiple groups and consortia such as the GLASS consortium active in this area. The question is an important one, since clinical therapeutic options at recurrence are limited, and a better understanding of how pathophysiology differs at recurrence would help to improve therapeutics.

The investigative goals of this study were only broadly defined - the two main goals of the study appeared to be 1) How tumour composition changes between primary and recurrent tumors 2) to use snRNA as a more comprehensive method of analyzing what types of cells play a role in tumour margins. **No clear hypothesis was defined a priori** - for example it was not made explicit what type of cell or process was hypothesised for being responsible at recurrence, and although it was stated that snRNAseq provided more detailed cellular compositional view due to its preservation of astrocytes and neurons, a potentially valid point, there was no hypothesis as to why investigation of these cells would be relevant and overarching goal seemed to be to provide a description of cell compositions.

Response: We appreciate the reviewer's comments and have more clearly stated the goals and hypotheses of the study in the revised manuscript. Our goal is to determine patterns of cellular composition and transcriptional states in primary and recurrent GBM, including both neoplastic glioma cells and non-neoplastic brain cells. We now state this in the first paragraph of the introduction. We focused our analysis on the non-neoplastic astrocytes, since these cells have been underrepresented in previous studies, and we identified a prognostically significant subpopulations of reactive astrocytes that shows phenotypic similarities to reactive astrocytes in other neurological diseases, such as neurodegenerative diseases. In the revised manuscript we have moved these finding to **Figure 1**. Our findings from the snRNAseq lead to 2 hypotheses; 1) The transcriptional landscape of GBM is determine by patterns of co-habitation of specific types and transcriptional states of neoplastic and non-neoplastic cells – now stated in the first paragraph of the second results section. 2) The tissue states defined by these patterns of cellular co-habitation are associated with metabolic dependencies – this is now stated in the first paragraph of the third results section. Data addressing the first hypothesis is provided in **Figures 2 and 3** of the revised manuscript. Data supporting the second hypothesis is included in **Figure 4** of the revised manuscript.

Comment: In terms of experimental approach, although snRNA seq does provide certain advantages - e.g. ability to analyze banked samples, it also has drawbacks, e.g. the loss of non-nuclear RNA information (which may or may not be relevant). Furthermore, although returning to frozen banks is a definitive

advantage, a key weakness of this study is the low number of matched primary/recurrent pairs, which the authors acknowledge. Without primary-recurrent pair testing, and given the heterogeneity of the TME, comparing a series of distinct primaries with distinct recurrences loses some of its value. Furthermore, one of the explicit goals is to explore the nature and cellular composition of margin tissue. Yet information as to the spatial location of the samples is not made clear from the methods (were these marginal tissue samples?). Even in Supplementary Table 1, it is not clear what the locational provenance of the samples analyzed is. This weakness was partially remedied through spatial analysis and re-analysis of previously collected bulk seq data with spatial information, however given the unclear provenance of the initial discovery dataset, any interpretation which transfer findings from the snRNA dataset must be treated with caution.

Response: We agree with the reviewer's assessment of the advantages and limitations of our snRNAseq dataset, and have revised our manuscript to focus on our findings regarding the patterns of co-habitation, and the 3-tissue state model. This approach is based on cellular composition analysis derived from snRNAseq from primary and recurrent GBM samples, including highly cellular tumor and infiltrated brain tissue. This is an unbiased approach that does not require knowing the spatial localization of the biopsies. We also derived tissue state gene signatures from the snRNAseq data, and used these signatures to assess the distribution and abundance of the tissue states in bulk RNAseq data from existing datasets of paired primary and recurrent GBM (**Figure 3**) and MRI-localized biopsies (**supplementary Figure s12**).

Comment: In terms of patient selection, lower grade gliomas were included, which consisted of two oligodendrogliomas, and an anaplastic astrocytoma; the rationale for including these in the analysis is again not clear as oligos represent quite a distinct pathological entity, and having IDH mutant primaries without recurrences again makes comparison and interpretation difficult.

Response: We agree with the reviewer that oligodendrogliomas and IDH-mutant astrocytoma are distinct pathological entities. We have revised the manuscript to more clearly explain that our rationale for including these samples in our study is "to include a spectrum of neurological diseases with alterations to non-neoplastic cells in the brain microenvironment", this is now included in the first paragraph of the first results section.

Comment: The finding that recurrent tumours have a more mesenchymal state is interesting and corroborates what others such as Wang et al. have noted. Indeed the proneural to mesenchymal shift with its attending changes in the immune environment can probably be seen as a defining feature of recurrence.

Response: We agree with reviewer and thank them on making this important point. Our findings that tissue state B is enriched in post-treatment recurrent tumors is consistent with those of previous studies, and further demonstrate how the patterns of cellular co-habitation correlate with this mesenchymal shift. In particular, our findings highlight the contributions of a specific subpopulation of nonneoplastic reactive astrocytes (Ast3) to the mesenchymal signature in GBM.

Comment: Astrocytes - the analysis of tumour associated astrocytes was an interesting investigation, as noted previously these cell types are infrequently studied. 9 clusters of these cells were identified from 707 cells. Such a low number of cells means that the significance of the marker genes from each cluster is limited. The identification of cluster 6 as one which was associated with prognosis was made. The genes CD44, CHI3L1, LGALS3, CLU and APOE were identified as genes marking this cluster, however no attempt was made to understand how these genes relate to each other and why they would have prognostic relevance. Indeed, having some form of astrocytic gene signature correlate with poor outcome in astrocytomas is not surprising, but it would be useful to know whether this prognostic relationship has any basis in biological studies or tissue localisation. Importantly a more systematic comparison to normal astrocytes and the heterogeneity of the latter, would have been valuable.

Response: We appreciate the reviewer's comments and agree that the prognostic significance of the astrocyte signatures merits further investigation, and should be put into context with other studies. To address these points, we consolidated the sub-clusters of non-neoplastic astrocytes into three main astrocyte states based on enrichment of protoplasmic and reactive astrocyte gene signatures and performed a comparison between these 3 major groups of glioma associated astrocytes and astrocytes we previously described in control brains and Huntington disease (Al-Dalahmah et al 2020). Our results showed that one of the subpopulations of astrocytes (Ast1) most closely resembles astrocytes from control brain, whereas another subpopulation of astrocytes (Ast3), most closely resembled astrocytes from Huntington disease. We further show that the Ast3 gene signature is highly enriched in prognostically relevant genes CD44, CHI3L1, LGALS3, CLU and APOE. This analysis is provided in our revised manuscript (**Figure 1 and supplementary table 4**). We moved the discussion of the astrocyte subclusters into the supplementary results.

To further address the reviewer's point on exploring how CD44, CHI3L1, LGALS3, and CLU gene expression is coordinated in astrocytes and how it relates to prognosis, we have conducted functional experiments to begin investigating these important questions. We transduced human astrocytes cultured in vitro with CLU-GFP, Gal-3-GFP, or control GFP lentiviruses before co-culture with U87 glioma cells for 24 hours. Next we FACsorted the cells and measured the expression of select genes using RT-qPCR – **Additional figure 1A**. We found that co-culture of astrocytes with glioma cells altered astrocytic gene expression in multiple ways, and that increasing astrocytic CLU or Gal-3 led to a significant increase in

CHI3L1 expression, suggesting their expression may be coordinated – **additional figure 1B**. Glioma gene expression was also influenced by co-culture with astrocytes, including changes in genes involved in glioma stem cell phenotype (PMC2897968, PMC5630302) – **additional figure 1C**. While these complex

Additional figure 1. A) Cartoon depicting the experimental outline of astrocyte glioma co-culture. B-C) Relative expression of select genes in sorted astrocytes (HA – panel B) and glioma cells (U87 – panel C) normalized to GAPDH. The expression is further normalized to un-transduced control astrocytes or U87 cells not previously co-cultured (delta-delta-CT). Log delta-delta CT values are shown. Paired t test. N = 3 independent biological replicates. * denotes p value < 0.05 – two-tailed t tests. # indicates p value < 0.05 – one-tailed t-test. Red color of signs indicate comparison to control HA or U87 not in co-culture. Green signs denote comparison to GFP condition (HA in B and U87 in C).

findings are very interesting, we believe additional work is needed to elucidate the mechanism underlying astrocyte-glioma interaction, and connect it to the overall narrative of the manuscript. We feel that these experiments are beyond the scope of this manuscript. We choose to include this data in this letter and not the manuscript to keep the emphasis on the tissue state model, and the metabolic perturbation experiments (**Figure 4**).

Comment: The use of the re-convolution technique in the study seemed like a superfluous exercise to a good extent. As far as I can make out the process is the reconstruction of sample cellular composition by aggregating all the cellular states and splitting them using dimensionality reduction to find 3 predominant "tissue states" across samples. These tissue states describe a cell type grouping which can be found across samples and correspond to infiltration, reactive changes and proliferation. How does this differ from just looking at the sample cellular composition using immunohistochemistry? One advantage may be to better define the cellular composition characteristics and the cross-talk and interactions which occur within each of these tissue states. However, this was not explored. Tissue states were correlated with prognosis, and again the finding that a mesenchymal signature is associated with worsened prognosis does not put us any further than we were before.

Response: We have revised the manuscript to better highlight what we gained from the tissue state analysis. Primarily, the reconvolution of the samples provided a novel and informative way to reduce the dimensionality of the snRNAseq data. The 3 tissue state model which resulted from this analysis fits well with neuropathological features seen in both primary and recurrent GBM samples, and provides gene signatures that can be applied to bulk RNAseq datasets, to reveal transcriptional patterns that correlate

with the underlying patterns of cellular composition. These points being made, we respectfully disagree with the reviewer that the value of the tissue state model does not exceed “looking at sample cellular composition using immunohistochemistry”. We used IHC as a validation of the signatures derived from the RNAseq rather than a means to define the tissue states. The RNAseq data is clearly more rich with information, and allowed us to discover additional functional and metabolic signatures that allowed us to extend this study to novel findings based on cellular co-habitation.

Comment: Enrichment of metabolic pathways was analyzed, and again although potentially interesting leads were highlighted, such as the glutathione pathway, none were explored even preliminarily.

Response: We agree with the reviewer that enrichment of metabolic pathways in tissue states is very interesting. We therefore explored this further both in vitro and ex vivo by focusing on the fatty acid synthesis pathway. We provide the results of these new experiments in **Figure 4** of the revised manuscript.

Conclusion

The use of snRNA from frozen tissue and tissue banks is an important technical advance which allows study of longitudinal samples and interrogation of existing biobanks, and the authors should be commended on obtaining data of sufficient quality to perform detailed analyses. However the study overall suffers from a lack of focus - if the aim of the study was to study primary and recurrent differences then matched pairs would provide for a better experimental design and tissue substrate. If however the goal was to study margin tissue, then a more targeted approach in tissue collection, with specific reference to how and where tissue was sampled from would be better suited to answering this question. Furthermore methods such as re-convolving and assembling tissue states by clustering of cellular states does not give additional information beyond what is already seen and observed using standard immunohistochemistry. The observation that certain cell types tend to cluster together in tumours into infiltrating regions, reactive regions and cellular regions is not a novel finding, and although confirmed with spatial studies it does not get us any further in terms of understanding how these cell types interact to cause disease or recurrence. This study represents a somewhat superficial descriptive study of an important area, and the findings and the techniques employed do not advance our understanding in an appreciable way.

Response: We believe that the revised manuscript clearly describes the investigative goals, is more focused, and features a coherent story built around the concept of co-habitation of specific cell-types with specific transcriptional states. We have validated this concept using both a compositional approach and a more extensive spatial transcriptomics approach. We focused on non-neoplastic astrocytes in the GBM microenvironment and emphasized that understanding cellular co-habitation in tissue states is both prognostically and functionally important. We demonstrate how a prognostically important tissue state can

be targeted in vivo by blocking metabolic functions of the cells that constitute it, namely astrocytes. Our re-convolution approach is now backed by experiments that offer a deeper understanding of the functional and prognostic relevance of co-habitation of cell-types in tissue states. Overall, our study provides novel insights and significant advances our understanding of reactive astrocytes may promote glioma progression.

Reviewer #2 comments:

Comment: The sample size is somewhat low, if the authors can increase that it would improve their power.

Response: We appreciate the reviewer's comment on increasing the n of our study. Having analyzed more than 35,000 nuclei from 21 donors, and 91 novel bulk RNAseq samples, we believe our data would be better served by providing orthogonal analysis using a different approach- spatial transcriptomics. Therefore, we expanded our study to include spatial transcriptomics analysis of 9 additional GBM samples.

Comment: The authors have validated their findings using 91 published primary and recurrent GBM bulk RNA-seq datasets. However, there are several hundred bulk GBM datasets at least which contain primary and recurrent specimens in the literature. I would ask the authors to increase their sample size. Some power analysis would increase confidence. The authors have mostly used the bulk RNA-seq to confirm the microenvironment gene expression. I would ask the authors to also confirm some of their other findings in bulk RNA-seq data.

Response: Per reviewer's suggestions, we have extended our analysis to the following previously published datasets: Wang L. et al. 2021, which includes bulk RNAseq data from paired primary and recurrent GBM (**Figure 3**), and Gill B., et al 2014, which includes bulk RNAseq (**supplementary Figure S12**)

Comment: Please provide QC metrics on the snRNA-seq in the supplement, including the distribution of number of genes tagged and doublet rates. Along these lines, did the authors remove doublets? They should if they haven't.

Response: We now provide the QC requested by the reviewer in supplementary table 1. Note, we loaded only 5000 cells per sample to reduce the doublet rate. The doublet rate for the samples loaded is 2.3%. Based on the recovered cells, the doublet rates range from 0.4-3.9%. The details are provided in the methods section under snRNAseq analysis - data cleanup.

Comment: The mesenchymal and proneural signatures don't seem to contain canonical markers. Please plot some more standard markers (CD44, CHI3L1 for mes, and OLIG2, ASCL1 for proneural for example).

Response: The conventional markers are provided in supplementary tables 2 and 3 (top markers sheet), and confirm the expression of mesenchymal markers CHI3L1 and CD44 in gl_mes1/2, and canonical proneural markers (SOX4, CD24, STMN1, GRIA2, OLIG1) in gl_PN1/gl_PN2. Per the reviewer's request, we provide some of these markers plotted in **supplementary Figure S4, panel I**. The results confirm that our glioma states indeed fall along the classes previously published and are presented in **supplementary Figures S1E and S3F** of the revised manuscript. Moreover, gene-set enrichment analysis of entire genesets based on Verhaak et al, Neftel et al. and others confirm these findings - (**now provided in supplementary Figures S1D and S3D of the revised manuscript**).

Comment: Moreover, it is not clear why the proliferative state is separate from these other states. Previous studies show that both the proneural and mesenchymal cell types proliferate. Can the authors compute a proliferation score say via Seurat and score their other cells to see which compartments are cycling.

Response: We provide additional analysis to address this point. Scoring cells by Seurat's method shows that indeed all glioma states display a proportion of cycling cells in G2M and S phases. Only gl_Pro1 and gl_Pro2 are entirely cycling (provided in **new supplementary Figure S4, panel I**). This is now added to the fourth paragraph of the first section of the supplementary results.

Comment: The microenvironment signatures from astrocytes are interesting but no validation has been done that these factors are elaborated by neoplastic cells or astrocytes, or that they promote malignant progression or are therapeutically relevant beyond correlation. Any work done here increase the significance of the manuscript and any novel targets and demonstration of being able to modulate viability or survival by targeting these signals would increase the value of the manuscript.

Response: We agree that demonstrating the functional significance of our analysis would increase the impact of the manuscript. To address this point, we have now included a perturbative study using acute slices treated with treated with FASN inhibitor to test the effects of targeting a metabolic signature associated with tissue state B. This new data is included in **Figure 4** of the revised manuscript.

Comment: Very little has been done to integrate the plate-seq data and snRNA-seq from the same patient. Moreover, the plate-seq data is a small sample size (n=1). As it stands this data does not really add much since it is underpowered and yet not integrated with the other data in the manuscript.

Response: We chose to integrate the snRNAseq with the plateseq data by clustering the latter on the enrichment scores of the compositional signatures derived from the former. This in essence allowed to build our three tissue state model. Moreover, to the reviewer's point, we now added 9 new spatial transcriptomics datasets that we use to integrate with the results from snRNAseq and orthogonal evidence for the co-habitation of the cell-types/states that make up our three tissue states. The new results are now presented in **Figure 2, panels D-E** of the revised manuscript.

Comment: The fonts on most of the figure panels are way too small to read, especially the GO lists and gene lists on the heatmaps. It is not necessary to provide a long list of GO terms in a figure. This can be in a supplementary table if necessary. Either way, the figure quality needs to be greatly improved.

Response: We now enhanced the quality of the figures to make the text easier to read. The full lists of GOs are provided in the **supplementary tables (S2-4 and S6)**.

Comment: The survival analyses in Figure 4-5 are not controlled for age and sex, therefore should be viewed as preliminary. Also, it is not clear if they are controlled for IDH-mutation status which is critical.

Response: Cox proportional hazard modeling was performed to control for age, gender and MGMT status in IDH-WT GBM. We provide this data in **Figure 3** in the revised manuscript. Note, only IDH-WT cases are included in the analysis. The data reveals that enrichment of tissue state B signature is associated with increased risk of death.

Reviewer #3 Comments:

Comment: The overall quality of the data is difficult to evaluate because of the lack of information on the average number and genes per cell and other quality characteristics.

Response: We provide quality metrics on the samples we analyzed as provided in **supplementary table-1** of the revised manuscript.

Comment: Malignant cells identified by clusters and/or InferCNV, etc., present another challenge. If the individual patients cannot be easily integrated (which is the case for most datasets), other analysis approaches must be taken (see Neftel et al.), and uniform transcription programs should be sought.

Response: We appreciate the important points the reviewer brings up. We completely agree that integrating tumor samples from different patients is challenging. We indeed found a strong patient-effect when we aligned the sequencing batches. That was the case for transformed CNV positive cells, but not

the case for untransformed CNV negative cells. We therefore controlled for the patient in the `sctransform` formula in Seurat 3, which employs canonical correlation analysis and *mutual nearest neighbors*-anchors. This method was amongst the top performing methods in batch integration as described in this thorough analysis of computational methods used for data integration (<https://doi.org/10.1186/s13059-019-1850-9>). This allowed us to cluster glioma cells and discover their heterogeneity in a manner that recapitulates the results of Neftel et al. 2020, and derive gene signatures we externally validated.

Comment: Here, the presentation of the data lacks information about the patients in each identified cluster. For UMAP, I would strongly suspect that the identified groups are primarily patient-specific (Figure 1).

Response: We hope to make this information more easily accessible. In the revised manuscript, the sample-by-sample composition of each cluster, is provided in **supplementary table-1** and illustrated graphically in **Figure 2** of the revised manuscript. It is provided for convenience in the additional table below. These results demonstrate that glioma clusters are not patient-specific factors.

case	gl_Mes1	gl_Mes2	gl_PN1	gl_PN2	gl_Pro1	gl_Pro2
TB8762	362	148	2	429	19	141
PA1	933	115	462	85	42	88
PA2	373	38	185	12	17	50
PA3	409	128	1037	1310	30	113
PO2_c	442	117	87	20	66	127
PO2_m	6	0	11	3	1	2
TB3864	281	29	0	56	21	77
TB3966	224	42	1	499	10	33
TB4027	148	77	0	0	5	11
TB4416	19	32	0	0	1	2
TB4718	199	16	75	10	14	29
TB4849	1592	339	1	11	98	301
TB4916	436	59	155	625	9	18
TB5053	21	16	234	7	0	0
TB5124	1231	295	563	366	40	105

Comment: The yield of the experiments sounds very low: 15189 nuc from 8 samples when using the 10X system seems very little to me. How did this happen? were several patients loaded per lane? How many nuc. were loaded per lane? The expected number would be between 60tds and 80tds nucs.

Response: In order to reduce the rate of doublets, we submitted 5000 nuclei per sample, and loaded multiple samples on the same sequencing lane – to minimize sequencing batches. We then applied stringent QC parameters to remove incompletely homogenized nuclei by filtering events with more than 75-80% exonic reads – which are unlikely to be nuclear (PMC6306246). Reducing the doublet rate is essential to be confident about the results of chromosomal copy number alteration analysis. Overall, the number of cells per sample is within what we normally get for other projects. However, some of the samples that we

used came from limited patient material, biopsies that are roughly 0.5*0.2*0.1 cm overall, which gave rise to low number of nuclei salvaged.

Comment: In the first part, the authors identified transcriptional subtypes that largely overlap with known states. Direct comparison with the established classification (Neftel) is lacking. The authors should also consider using the Neftel states as a reference map, as the main goal of this work is not a novel single-cell classification.

Response: We agree with the reviewer's comment that our goal is not to generate a novel single-cell classification of glioma cells. Rather, we want to use what is already known about glioblastoma to facilitate identification and classification of glioblastoma cells. To that end, we have provided direct comparisons between our glioma states and those described by Neftel et al., as well as others such as Levitin et al. and Wang et al (Supplementary Figure S1 and S3). We also appreciate the reviewer's suggestion to use Neftel states as a reference map. To this end, we used genesets derived from the aforementioned papers as "roadmaps" and directly measured their enrichment in our dataset. This is a direct form of comparison taking a conclusion from one paper "genesets" and testing its enrichment in another.

Comment: From the H&E and the CNV profiles, the margin samples seem to cover different areas from infiltrating areas to normal brain (PO1). How were the samples guided and planned are the points saved in the Neuronavigation system? It would be interesting to show the distance to the contrast enhancing region for each "margin" sample.

Response: We agree with the reviewer, annotating samples with objective measurements taken at the time of sample acquisition using the NeuroVavigation system would provide additional information. Unfortunately, the samples used for snRNAseq were not radiologically localized or mapped in relation to an objective landmark. While this idea is very enticing, accomplishing it will require adding a new cohort of samples to be acquired prospectively. This falls beyond the scope of this manuscript,

Comment: The authors showed that similar states and state distributions were observed in primary and recurrent samples. Why not integrate both types of data and show the similarity?

Response: Per reviewer's suggestion, we integrated the primary and recurrent glioblastoma cells using CCA+MNN alignment as per Seurat 3. The analysis shows that the cells from primary and recurrent samples overlap in the UMAP space, and that this overlap is seen for all 6 GBM states (Presented in **supplementary Figure 5G**). Additional analysis demonstrated how glioma states are correlated in expression of the marker genes (**supplementary Figure S3**).

Comment: The concept of spatial heterogeneity and microenvironmental crosstalk sounds interesting but is not explored further. The addition of spatial transcriptomics with single cell integration of the snRNA-seq dataset (Cell2Location/Spotlight) will add significant value.

Response: To address the reviewer's comment, we added 9 additional spatial transcriptomics (ST) datasets. As means to integrate snRNAseq and ST datasets, we adopted an approach parallel to that outlined in Cell2Location/spotlight analysis, and advocated for by the authors of the Seurat package, and aimed for studying the scores of specific snRNAseq-derived genesets. We calculated the enrichment of cell-type and cell-state gene signatures for all point in the ST dataset and projected an example in panel **Figure 2D (new panel)**. The results show how signatures we predict to co-inhabit the same sample indeed are spatially adjacent (Ast3 and gl_Mes2 for example). We next established the concept of co-habitation between cell-types/cell-states in the spatial landscape using correlation analysis of cell-type derived sample signatures (**Figure 2E**).

Comment: The greatest advantage of the manuscript is the microenvironment data. It is a pity that these data are not presented more concisely and that the focus is on microenvironment changes between primary and relapse status. The crosstalk of myeloid and glial cells plays a crucial role and should be more clearly elaborated here. Extrapolation of signatures to the TCGA database is not very helpful and does not provide new insights.

Response: We agree with the reviewer that the strength of our data is the focus on the tumor microenvironment, and the revised manuscript has been reorganized to focus on the tumor microenvironment – particularly non-neoplastic astrocytes. We agree that the interaction between myeloid and other glial cells is of great interest. We provide a panel showing some analysis using Cell Chat to highlight one such example of probable interaction between astrocytes and myeloid cells **Additional figure 2**. While this results points to complement signaling as a potentially target to disrupt crosstalk between astrocytes and myeloid cells in the GBM microenvironment, we feel that functional studies to further explore this interesting and complicated system is beyond the scope of the paper. Therefore, we have included these results in our letter, but have not added them to the paper.

As

Additional Figure 2. CellChat based cell-cell interaction network of the complement pathway, incriminating myeloid cells and various other cells including astrocytes. Expression of C3 and C4 is prominent in astrocytes, while that of the receptors is mainly in immune cells. another way to get at this issue, we investigated potential metabolic co-dependencies in the GBM microenvironment. We showed that tissue state B is enriched in metabolic pathways relating to fatty acid and lipid metabolism. Given that astrocytes are the primary cells in the brain that regulate lipid uptake through the blood brain barrier and metabolize fatty acids, we decided to test the effects of targeting the fatty acid synthase pathway. We hypothesized that blocking FASN will lead to depletion of the tissue state B transcriptional signature. Indeed, treating GBM explants ex vivo with FASN inhibitor Cerulenin depleted the tissue state B signature. This new data is included in **Figure 4** of the revised manuscript.

Comment: Direct crosstalk between glial and myeloid cells causing reactive transformation can be explored in the context of disease status. CellChat or other tools can be used to study the cellular interactions. Cellular evolution from a healthy cell to different activation states should be studied using RNA velocity or other tools to reconstruct the evolution of cell lines. Unfortunately, the authors did not use most of the established tools that help in the in-depth study of the datasets.

Response: We addressed the issue of crosstalk between cell types using CellChat in our response to the previous comment (see above). Please see additional figure 1 attached to this letter in our response to reviewer #1. Regarding the reviewer’s suggestion to study the cellular evolution from healthy to different activation states, this is an interesting idea, but there are technical challenges to applying RNA velocity to snRNAseq data, and further validation of this approach is needed (10.15252/msb.202110282). Therefore, we feel that this is beyond the scope of this paper.

Comment: The last part aims to examine tissue states that may correlate with clinical parameters. The concept does not become clear to me and cannot be addressed with the data shown.

Response: We regret that this the part of the manuscript was not clear in our initial submission and have revised the manuscript to more clearly explain the concept of tissue states, and to highlight the prognostic and therapeutic significance. We show that there are three compositionally-based transcriptional tissue states that summarize glioma infiltrated tissue, which derive from the co-habitation of cell types/states. These tissue states are prognostically relevant, as we show one of these tissue states – Tissue state B, is associated with increased risk of death in the TCGA-CGGA IDH-WT GMB dataset (**Figure 3H**). We demonstrate that co-habitation of cells in a tissue state is reflected in distinct metabolic signatures, which we show are functionally targetable via ex vivo treatment of GBM explant cultures with a FASN inhibitor (**Figure 4**).

REVIEWER COMMENTS

Reviewer #1 (Remarks to the Author):

The authors have re-submitted their manuscript regarding composition of the glioma tumor microenvironment in primary and recurrent tumors, with a focus on the non-neoplastic astrocyte population and expansion on their analysis of tissue states.

The authors have addressed many of the points raised, specifically they have reconfigured the study to delve into the role of astrocytes within the study. Although overall they have a reasonable sample size, (16,831 CNVneg nuclei assessed), the astrocyte population, which is now a major focus and the primary novelty of this study, is based on analysis of only 707 nuclei and only ~250 nuclei from recurrent or primary tumors. Although the concept of sample size is still very much a fluid target in single cells studies, making concrete conclusions on the basis of 707 cells seems tenuous. Additional data obtained either experimentally or gleaned from public datasets would have been helpful to bolster the case.

Nevertheless if we follow the line of reasoning, the authors identify 8 astrocyte sub-clusters based on non-biased sequencing analysis. However, these sub-clusters are then combined into “three astrocyte states”. A Huntington’s disease (HD) classifier is then applied to the astrocytes, however the reasoning for using this method, and using this disease (HD) as a comparison is unclear. Indeed, inspection of the original paper demonstrates that the original classifier was based on only 1064 cells in 6 HD patients and 6 controls. (Note: The description of the classifier in the manuscript is incomplete in the methods section). If Ast3 is similar to the HD group, then how similar? Could these datasets be integrated?

Even if we accept that the Ast3 cluster resembles astrocytes from Huntington’s disease, what would be the significance of this? No commentary is made on what links these disease processes. How do these astrocytes compare to other inflammatory or neurodegenerative diseases, what are the implications of the similarities or differences?

Astrocyte subcluster 6 appears to correlate most with prognosis (Figure S9) and more investigation into this cluster may be warranted if sample size can be increased, but this is unexplored in the manuscript.

The authors use spatial transcriptomics and identify spatial co-localization of cell types/transcriptional states within tumors. These findings are interesting though largely descriptive. The author’s revision/response to reviewers does provide some data investigating the interaction between glioma

cells and cultured astrocytes (Additional figure 1 and additional figure 2), but consider further exploration to be beyond the scope of this article.

The authors correlate patient prognosis with identified “Tissue states”. Given that it has been previously demonstrated that mesenchymal tumor states and specific monocyte-derived macrophage clusters correlate with poor prognosis, the novelty of the authors finding is limited since these are likely driving the Tissue state prognosis. The authors do not demonstrate that the addition of astrocyte state in the analysis has significantly improved prediction of prognosis over any prior analyses. With the low number of non-neoplastic astrocytes present within the tumors relative to the myeloid or tumor cells, it is also unclear how these astrocytes are significantly contributing to the prognosis.

The authors interestingly suggest that metabolic dependencies may be shared between cell types within the same tissue state. However, the investigation of these metabolic dependencies is superficial. The authors identify fatty acid synthesis as an enriched pathway in tissue state B, both in neoplastic and non-neoplastic astrocytes. By inhibiting fatty acid synthesis the authors find a depletion of tissue state B signature, but this argument is somewhat circular since much of this signature is likely based on fatty acid synthesis. Demonstration of differences in cell viability, cell-type composition or co-habitation, transition to other cell/transcriptional states would strengthen the case that fatty acid synthesis plays a critical role in the viability and composition of Tissue State B, and the resultant post-FASN inhibition phenotype and cellular composition would be of interest from a therapeutic and translational standpoint. Demonstration that depletion of specific states or state dependent metabolic pathways has a causal effect on outcome, whether it be in an in vivo or explant setting would strengthen the case that tissue states are important in tumor progression.

Overall the authors have responded to previous comments and provided a more focused manuscript highlighting the astrocytes with more explanatory basis for defining tissue states. Weaknesses as to the significance of the findings persist. The discussion should include a more pointed acknowledgment of low cell numbers (esp. for the astrocytes) and the relevance of similarity to neurodegenerative microenvironments.

Reviewer #2 (Remarks to the Author):

The authors have addressed most of my technical concerns from the first manuscript. The limited sample size has not been addressed, the manuscript is largely descriptive, likewise the novel spatial transcriptomics data added during revisions is interesting but also descriptive.

Reviewer #3 (Remarks to the Author):

My comments have been addressed to my satisfaction, but unfortunately I have some concerns about the new data that have been incorporated into the manuscript. In the current presentation, the spatial transcriptomic data are insufficient and the analysis is flawed in parts. I therefore urge you to revise the analysis and presentation of the data before publication. Here are some points that should help improve the spatial data analysis.

1. it is not clear to me where the samples came from and what they show. Is there no H&E staining to classify the origin histologically?
2. CNV analysis of the spatial data would help to understand if and where the tumor is localized in the samples, cluster analysis would help to classify the individual niches of the tumor.
3. How different are the regions? Comparing the 9 datasets in terms of similar transcriptional niches would help to better classify the data.
4. correlation analyses (Fig) are highly biased in spatial data by the false assumption that the individual spots are independent, which is not the case. Geographically weighted correlations or at least Monte Carlo simulations with random data should be performed to confirm correlation.
5. integration by gene set is not helpful because often different cells (astro and tumor), neurons and neuronal differentiation (tumor) or proliferating TAMs and tumor cells have high overlap of signatures. The last state of the art tool, RCTD, for example, helps to achieve better integration/deconv. of snRNA-seq data in this context.
5. The analysis of samples by region (k=10) seems to me flawed. What was the exact rationale for the step: "To increase the robustness of the downstream analysis, counts from ~20 adjacent ST spots were summed to 10 ST regions..." Why do the authors assume that random neighbors belong together and represent distinct spatial niches? It is also not clear to me why robustness is strengthened in this way. In other words, the analysis resembles the approach to average 20 random cells. Why are spots that are similar and belong to the same spatial niche not clustered together based on their cartesian and transcriptional similarity (e.g. BayesSpace clustering). And then used for downstream analysis...
6. Please add QC from spatial data, including UMI counts (also for snRNA-seq)

Again, the approach to generate and integrate spatial data is highly appreciated, but the analyses chosen here seem insufficient.

Reviewer 1 Comments

Comment: The authors have re-submitted their manuscript regarding composition of the glioma tumor microenvironment in primary and recurrent tumors, with a focus on the non-neoplastic astrocyte population and expansion on their analysis of tissue states. The authors have addressed many of the points raised, specifically they have reconfigured the study to delve into the role of astrocytes within the study. Although overall they have a reasonable sample size, (16,831 CNVneg nuclei assessed), the astrocyte population, which is now a major focus and the primary novelty of this study, is based on analysis of only 707 nuclei and only ~250 nuclei from recurrent or primary tumors. Although the concept of sample size is still very much a fluid target in single cells studies, making concrete conclusions on the basis of 707 cells seems tenuous. Additional data obtained either experimentally or gleaned from public datasets would have been helpful to bolster the case.

Response: We appreciate the reviewer's careful reading of our manuscript and helpful comments. To address the reviewer's concerns regarding astrocyte numbers, we have now included snRNAseq analysis from 7 additional GBM samples, which we treat as a validation set. With this new data, we have added 4635 CNV negative nuclei, including 1791 astrocytic nuclei, bringing our total astrocyte number to 2489 nuclei. This new data is now included in new supplementary figures S8 and S13A-C of the revised manuscript.

Comment: Nevertheless if we follow the line of reasoning, the authors identify 8 astrocyte sub-clusters based on non-biased sequencing analysis. However, these sub-clusters are then combined into "three astrocyte states". A Huntington's disease (HD) classifier is then applied to the astrocytes, however the reasoning for using this method, and using this disease (HD) as a comparison is unclear. Indeed, inspection of the original paper demonstrates that the original classifier was based on only 1064 cells in 6 HD patients and 6 controls. (Note: The description of the classifier in the manuscript is incomplete in the methods section). If Ast3 is similar to the HD group, then how similar? Could these datasets be integrated?

Response: We understand and appreciate the reviewer's critique. To address these comments we have revised our approach to classifying the astrocytes and have reexamined their relationship to astrocytes seen in neurodegenerative diseases. The specifics points of our response are provided below:

1) We have removed the sub-cluster analysis, and we performed a new analysis to cluster the astrocytes based on gene set enrichment analysis using curated gene signatures for three astrocyte phenotypes. This new analysis is presented in new **Figure 1** and the methods are described in the supplementary results section of the revised manuscript.

2) Given that we added new astrocyte data, we performed an additional analysis where we performed unbiased clustering on all of our astrocytes (including the new dataset) using the Louvain methods on shared nearest neighbor graphs generated in igraph. This analysis identified three clusters of astrocytes that resemble the three astrocyte states based on the astrocyte markers (**additional figure-1 in this letter, and Figure S13A-B in the revised manuscript**),

Additional Figure 1 - unbiased clustering of astrocytes A) UMAP plots B) marker expression dotplots

3) We then performed a comparison between assignments of Louvain clusters and Astrocyte states which shows that Louv_2 cells were mostly Ast1, Louv_3 were mostly Ast2, and Louv_2 were mostly Ast3- (additional figure-2 in this letter, and new supplementary Figure S13C in the revised manuscript).

Additional Figure 2 Scaled proportions of Louvain cluster assignments in each astrocyte state – scaled by column.

4- We also performed new analyses integrating glioma-associated astrocytes with astrocytes from Alzheimer’s disease, Huntington Disease and control post-mortem brains (additional figure 3 in this letter). We first reduced the dimension using PHATE (doi 10.1038/s41587-019-0336-3), then projected the cells in the shared reduced space. We corrected for batch effects using the mutual nearest neighbor method in PHATE. The results show that Ast1, and some control and ast2 astrocytes lie on the periphery, while Ast3 appear to be in the center of the graph with HD and AD astrocytes lying in between (Additional figure 3A). To further examine the potential relationship with neurodegeneration, we performed a linear discriminant analysis based classification using scID (Boufeau et al. 2022 DOI: 10.1016/j.isci.2020.100914) which showed that many AD astrocytes are unclassified (additional Figure 3B). Furthermore, this new analysis showed that most Ast1 resemble control astrocytes, most Ast2 resemble HD astrocytes, and Ast3 showed only a modest resemblance to either AD or HD astrocytes (additional Figure 3B). Notably, this result is different from what was seen in our previous analysis, which showed that Ast3 is most similar to the HD astrocytes. Therefore, we have revised our discussion on the potential relationship between astrocytes in glioma and neurodegenerative diseases and now limit

our analysis only to the results of differential gene expression analysis shown in **Figure 1G-H** of the revised manuscript.

Additional Figure 3 A) integration of glioma-associated astrocytes with neurodegenerative astrocytes projected in PHATE dimensionality reduced dimension space. B) LDA classification of neurodegenerative and control astrocytes by glioma-astrocyte states – scaled by column. Alzheimer’s disease astrocytes are from (Grubman et al. 2019 - DOI: 10.1038/s41593-019-0539-4), HD astrocytes from Al Dalahmah et al., 2021.)

Comment: Even if we accept that the Ast3 cluster resembles astrocytes from Huntington’s disease, what would be the significance of this?

Response: As explained above, we have now limited our comparison to neurodegenerative astrocytes to focus on the differentially gene expression analysis shown in **Figure 1G-H**. Our claims that Ast3 resembles astrocytes in Huntington’s disease has been removed from the revised manuscript.

Comment: No commentary is made on what links these disease processes. How do these astrocytes compare to other inflammatory or neurodegenerative diseases, what are the implications of the similarities or differences?

Response: We now include a commentary on the potential links between glioma-associated astrocyte gene expression and genes involved in neurodegenerative diseases. We specifically focus on CLU, a gene relevant to both Alzheimer’s disease and GBM, and we provide functional data based on astrocyte-glioma cell coculture experiments (**new supplementary figure S13**) and link to fatty acid metabolism, which is relevant to several neurodegenerative diseases. Identifying shared and dissimilar features between astrocytes and glioma expands our understanding of astrocyte pathology across different neurologic diseases.

Comment: Astrocyte subcluster 6 appears to correlate most with prognosis (Figure S9) and more investigation into this cluster may be warranted if sample size can be increased, but this is unexplored in the manuscript.

Response: We now removed the discussion of astrocyte subclusters from the manuscript, as noted in our response to point 1. Given the similarities between sub-cluster 6 (previously) and Ast3, we now include the functional data showing on CLU, a gene highly expressed in Ast3. We show that overexpression of CLU leads to increased expression of several other marker genes of Ast3, including CHI3L1, ID3, IGFBP7, TNC, and TGFBR2 (**new supplementary Figure S13**), and leads to changes in

the gene expression of U87 glioma cells causing upregulation of genes involved in proliferation, glial differentiation, and fatty acid synthesis.

Comment: The authors use spatial transcriptomics and identify spatial co-localization of cell types/transcriptional states within tumors. These findings are interesting though largely descriptive.

Response: We have revised and expanded our analysis of the spatial transcriptomic data, including a new analysis that further demonstrates the spatial co-localization of specific glioma cell states with specific non-neoplastic cell types. Of particular relevance to this manuscript, our spatial cross correlation analysis shows that Ast3 astrocytes colocalize with Mes2 glioma cells, and to a lesser extent, Mes1 glioma cells. More details about these new analyses are provided in our response to Reviewer 3.

Comment: The author's revision/response to reviewers does provide some data investigating the interaction between glioma cells and cultured astrocytes (Additional figure 1 and additional figure 2), but consider further exploration to be beyond the scope of this article.

Response: As suggested by the reviewer, we now include the co-culture studies in our revised manuscript (**new supplementary Figure S13**). This new data shows that CLU overexpression leads to increased expression of many of the genes that are correlated with high CLU expression (based on snRNAseq). Furthermore, co-culture of GFP or CLU overexpressing astrocytes with U87 glioma cells led to gene expression changes in glioma cells, and CLU overexpression led to upregulation of genes involved in glial differentiation, proliferation, and fatty acid synthesis.

Comment: The authors correlate patient prognosis with identified "Tissue states". Given that it has been previously demonstrated that mesenchymal tumor states and specific monocyte-derived macrophage clusters correlate with poor prognosis, the novelty of the authors finding is limited since these are likely driving the Tissue state prognosis.

Response: We would like to reiterate the major novel findings: 1) One major advance of our study over previous studies is that the snRNAseq provided a more comprehensive and unbiased sampling of the different cell types in the glioma microenvironment, and this enabled us to discover patterns of cohabitation that have not been previously described, including the cohabitation of Ast3 astrocytes and Mes2_glioma cells which is a major focus of our manuscript. We respectfully disagree that the novelty of our finding is limited. In fact, we contend that the cohabitation model is better able to capture the effects of multiple correlated cell types/transcriptional states that we show do indeed co-inhabit in glioma infiltrated tissue. Please also see below evidence in response to the point below that using the tissue state model is advantageous in predicting survival. Furthermore, the model allowed us to derive potential therapeutic targets that based on metabolic pathways not limited by one cell type, rather utilized by multiple cell types.

Comment: The authors do not demonstrate that the addition of astrocyte state in the analysis has significantly improved prediction of prognosis over any prior analyses.

Response: We now provide additional analysis of the combined CGGA-TCGA IDH-WT GBM survival series to address this point. (see additional figure 4 in this letter, and **figure 3H** in the revised manuscript). When corrected for age, sex, and MGMT status, we find the enrichment of the genesets for the individual cell types that compose tissue-state B do not reach significance. However, analysis of the composite tissue-state gene signature enrichment was significantly correlated with increased risk of death (**revised figure 3H**). These results demonstrate that accounting for multiple cell types can improve prediction of prognosis.

Comment: With the low number of non-neoplastic astrocytes present within the tumors relative to the myeloid or tumor cells, it is also unclear how these astrocytes are significantly contributing to the prognosis.

Response: We have now increased the number of astrocytes in our data, and our new results show that non-neoplastic astrocytes, particularly Ast3, are seen in relatively high abundance in some tissue samples, and that their abundance is significantly correlated with specific transcriptional states of glioma cells. These findings suggest that one way non-neoplastic astrocytes can contribute to prognosis is by affecting the transcriptional phenotype of glioma cells that they colocalize with in the GBM tissue. This idea is supported by the results of our functional studies showing that coculturing of astrocytes that resemble Ast3 with high CLU expression leads to transcriptional alterations of glioma cells, including upregulation of genes involved in glial differentiation, notch signaling, and proliferation such as SOX2, SOX4, HES1, and HES5 (**supplementary figure S13 and supplementary table 8**).

Comment: The authors interestingly suggest that metabolic dependencies may be shared between cell types within the same tissue state. However, the investigation of these metabolic dependencies is superficial. The authors identify fatty acid synthesis as an enriched pathway in tissue state B, both in neoplastic and non-neoplastic astrocytes. By inhibiting fatty acid synthesis the authors find a depletion of tissue state B signature, but this argument is somewhat circular since much of this signature is likely based on fatty acid synthesis.

Response: This is an important point. We have elaborated on the methods used to derive the tissue state signatures in the methods section, and rederived our signatures to increase the number of genes per signature. Our tissue state B has now 481 unique genes, as opposed to 94 genes previously. Even when we increase the number of gene in our signatures, almost none of the tissue state B signature genes are part of the fatty acid biosynthesis pathway. To further elaborate on this point, FASN is not in tissue state B gene signature. An inspection of the more general GO - cellular lipid metabolic process (GO: 0044255 n = 31 genes) shows that there are no genes shared with tissue state B gene signature. Another ontology, fatty acid biosynthesis pathway (GO ID: 0006633 – n = 75 genes), shares only one gene with the genes unique to tissue state B - PLA2G4A. Thus, there is minimum overlap between our Tissue state B signature and the fatty acid metabolism-related genes. This does not mean that Tissue state B "samples" do not show enrichment for specific metabolic gene signatures. Tissue state B samples were indeed enriched in genesets related to fatty acid metabolism, but these were not part of the tissue state B gene signature that we used to measure the enrichment as an "endpoint". Therefore, our findings show that blocking fatty acid synthesis resulted in transcriptional alteration that extend beyond genes that

Additional Figure 4 Survival analysis using cox-proportional hazard model correcting for age, sex, and MGMT.

Enrichment of genesets was calculated using ssGSEA.

are part of the fatty acid synthesis pathway, including causing a significant decrease in the tissue state B gene signature.

Comment: Demonstration of differences in cell viability, cell-type composition or co-habitation, transition to other cell/transcriptional states would strengthen the case that fatty acid synthesis plays a critical role in the viability and composition of Tissue State B, and the resultant post-FASN inhibition phenotype and cellular composition would be of interest from a therapeutic and translational standpoint. Demonstration that depletion of specific states or state dependent metabolic pathways has a causal effect on outcome, whether it be in an in vivo or explant setting would strengthen the case that tissue states are important in tumor progression.

Response: We agree with the reviewer that additional experiments to further explore the effects of FASN inhibition could lead to significant new insights and potentially point the way to new therapeutic strategies. However, conducting such experiments is a very large effort that involves collection of GBM surgical explants and additional snRNAseq. These studies will take over a year to conduct. Although we have begun preparation to do these experiments, we feel that they are beyond the scope of this study, which already has multiple snRNAseq, bulk RNAseq, spatial transcriptomic, IHC, and ISH datasets.

Comment: Overall the authors have responded to previous comments and provided a more focused manuscript highlighting the astrocytes with more explanatory basis for defining tissue states. Weaknesses as to the significance of the findings persist. The discussion should include a more pointed acknowledgment of low cell numbers (esp. for the astrocytes) and the relevance of similarity to neurodegenerative microenvironments.

Response: We believe we now focused the manuscript and addressed all the concerns raised by the reviewer. We have also added more astrocytes to our dataset by including seven additional snRNAseq samples. We have also revised our discussion on the similarities to neurodegenerative disease to focus on the functional relevance of CLU, a gene that is expressed by reactive astrocytes in both GBM and Alzheimer's disease. Furthermore, we have improved the discussion to address the points raised by the reviewer. We hope that the current version is to the reviewer's satisfaction.

Reviewer 2 Comments

Comment: The authors have addressed most of my technical concerns from the first manuscript. The limited sample size has not been addressed, the manuscript is largely descriptive, likewise the novel spatial transcriptomics data added during revisions is interesting but also descriptive.

Response: We have addressed the concerns of limited sample size by increasing the size of our snRNAseq data, which now includes 43,505 cells from 29 patient samples. The study also includes spatial transcriptomics from 9 samples and bulk RNA sequencing from 91 samples. We have also significantly revised and expanded our analysis of the spatial transcriptomic data to further characterize the colocalization of specific cell types and transcriptional states in GBM. We have also included additional perturbative studies including co-culture experiments showing that overexpressing the A23 signature gene Clu in astrocytes can affect the transcriptional phenotype of glioma cells. These co-culture experiments, along with the slice culture experiments included in the prior submission, provide new insights into the mechanisms that affect cellular composition and phenotypes within GBM tissue states.

Reviewer 3 Comments

Comment: My comments have been addressed to my satisfaction, but unfortunately I have some concerns about the new data that have been incorporated into the manuscript. In the current presentation, the spatial transcriptomic data are insufficient and the analysis is flawed in parts. I therefore urge you to

revise the analysis and presentation of the data before publication. Here are some points that should help improve the spatial data analysis.

Response: We appreciate the reviewer's suggestions and have revised our analysis and presentation of the spatial transcriptomics data to address the reviewers comments. A point by point response is provided below.

Comment 1: it is not clear to me where the samples came from and what they show. Is there no H&E staining to classify the origin histologically?

Response: We did not stain the tissue sections used for ST using H&E. Rather, we performed immunofluorescence on all sections used for ST to label neuronal nuclei with NeuN and total cell nuclei with DAPI. These two markers provide a map of the neuronal density and total cellularity of the tissue sections that we quantified and correlated with the ST data. This data is included in **new supplemental figures S10 and S11**. We also performed H&E stains on other sections from the same tissue blocks used for ST to confirm that all samples contained cellular tumor. However, it was not possible to make a direct spatial correlation between these H&E stains and the ST data. Therefore, we used the data from the immunofluorescence analysis that was performed on the ST sections.

Comment 2: CNV analysis of the spatial data would help to understand if and where the tumor is localized in the samples, cluster analysis would help to classify the individual niches of the tumor.

Response: To provide a spatial map of tumor burden, we obtained an estimate of the fractional proportion of CNV+ cells for each spot in each ST sample using deconvolution with RCTD, as suggested by the reviewer in point 5 below (**new Supplementary figure 11**). We also performed a correlation analysis between the tumor burden as inferred by deconvolution, and cellularity by quantifying DAPI+ nuclei in the ST sample sections. The results of the latter analysis showed a positive correlation between the two metrics (**Supplementary figure S11** and supplementary results section "Spatial cross-correlation analysis of deconvolved cell type proportions").

Comment 3: How different are the regions? Comparing the 9 datasets in terms of similar transcriptional niches would help to better classify the data.

Response: To address this comment, and the comment on cluster analysis, we performed BayesSpace analysis to investigate how regions within our ST samples differ from one another as suggested by reviewer. We then used the cell density measurement from the immunofluorescence analysis to calculate the average cell density for each BayesSpace cluster and calculated the correlation coefficient between that and the fractional composition of CNV+ tumor cells for each BayesSpace cluster, as derived by deconvolution. We found a significantly positive correlation between the two tumor burden measures – providing support for deconvolution as a measure to assess tumor burden. This analysis also shows that transcriptional regions, as defined by BayesSpace, can differ by tumor burden, or put another way, tumor burden contributes to the transcriptional landscape of GBM. However, the results of our deconvolution analysis also highlight that patterns of cellular abundance do not faithfully obey the boundaries defined by BayesSpace analysis. These findings are discussed in the supplementary results section of the revised manuscript.

Comment 4: correlation analyses (Fig) are highly biased in spatial data by the false assumption that the individual spots are independent, which is not the case. Geographically weighted correlations or at least Monte Carlo simulations with random data should be performed to confirm correlation.

Response: We acknowledge the reviewer's point and thank them for suggesting alternative approaches to the data. As per their suggestion, our revised analysis includes a geographically weighted correlation metric. Specifically, we used geographically weighted "cross correlation" analysis that quantifies the spatial cross correlation between our deconvolved cell types. We performed permutation testing to

confirm significant spatial correlations between cell types. The statistically significant spatial cross correlation patterns we observe identified common patterns of cohabitation between specific cell types across our data set. These findings are now included in **new Figure 2** of the revised manuscript.

Comment 5: integration by gene set is not helpful because often different cells (astro and tumor), neurons and neuronal differentiation (tumor) or proliferating TAMs and tumor cells have high overlap of signatures. The last state of the art tool, RCTD, for example, helps to achieve better integration/deconv. of snRNA-seq data in this context.

Response: We appreciate the reviewer's constructive suggestion and have revised our analysis accordingly. In our revised analysis, we applied their suggested technique, RCTD, to deconvolve the presence of cell types in our ST samples. These results are presented in new **Figure 2D** and **supplementary Figure 12** of the revised manuscript.

Comment 6: The analysis of samples by region (k=10) seems to me flawed. What was the exact rationale for the step: "To increase the robustness of the downstream analysis, counts from ~20 adjacent ST spots were summed to 10 ST regions..." Why do the authors assume that random neighbors belong together and represent distinct spatial niches? It is also not clear to me why robustness is strengthened in this way. In other words, the analysis resembles the approach to average 20 random cells. Why are spots that are similar and belong to the same spatial niche not clustered together based on their cartesian and transcriptional similarity (e.g. BayesSpace clustering). And then used for downstream analysis...

Response: To address these points, we used geographically-weighted correlation analysis of deconvolved proportion rather than GSEA (per reviewer's suggestion) as detailed in the supplementary results and as explained below: Using the deconvolved cell type proportions at each spot, we analyzed cohabitation patterns using spatial cross correlation, a mathematical approach that has been previously used to analyze spatial relationships in gene expression.^{1,2} Here, we implement spatial cross correlation to quantify and aggregate the spatial correlation of deconvolved proportions of transcriptionally-distinct cell types across our 9 ST data sets. Spatial cross correlation allows us to quantitatively assess the covariance of any two cell types, A and B, by comparing the predominance of cell type A in any given spot to the predominance of cell type B in that spot's geographic neighborhood. We calculate this metric for every spot and for every pairwise comparison of cell types across all of our ST samples. This approach strengthens the validity of our observed spatial cohabitation patterns because, as the reviewer noted, we cannot assume that spots in close proximity to each other are independent. This spatial cross correlation approach allowed us to discover patterns of cohabitation that we provide in **Revised Figure 2** - specifically, that we can capture patterns of spatial co-habitation similar to those we observe based on compositional analysis using our PCA analysis. As explained above, we also performed BayesSpace analysis (see response to reviewer comments 2 and 3).

Comment 7: Please add QC from spatial data, including UMI counts (also for snRNA-seq)

Response: We provide these QC metrics in **supplementary table 1**.

Comment 8: Again, the approach to generate and integrate spatial data is highly appreciated, but the analyses chosen here seem insufficient.

Response: We hope that the revised analyses address the reviewer's concerns.

References:

1. Chen Y. A new methodology of spatial cross-correlation analysis. *PLoS One*. 2015;10(5):e0126158. Published 2015 May 19. doi:10.1371/journal.pone.0126158
2. Miller BF, Bambah-Mukku D, Dulac C, Zhuang X, Fan J. Characterizing spatial gene expression heterogeneity in spatially resolved single-cell transcriptomic data with nonuniform cellular densities. *Genome Res*. 2021;31(10):1843-1855. doi:10.1101/gr.271288.120

REVIEWER COMMENTS

Reviewer #1 (Remarks to the Author):

The authors have re-submitted their manuscript and should be commended on the fact that many of the points raised have been addressed and the overall story is much more compelling.

Comment 1

The addition of the validation cohort strengthens the contention that distinct astrocyte subgroups exist in these sample cohorts and further defines the Ast 1-3 states. This in turn feeds into the cross validation with neurodegenerative diseases with Ast3 having more similarities to both AD and HD than just HD as previously described.

Perhaps I am mistaken but the link between CLU and Ast3 does not seem particularly strong. If anything CLU expression, which we are using as a surrogate markers for astrocytes associated with neurodegenerative diseases, is higher in Ast2 (see annotated figure below).

Indeed with CLU overexpression the increase of Ast3 associated genes such as CHI3L1, ID3 are also highly expressed in Ast2. Some explanation as to why CLU overexpression is felt to be more representative of Ast3 and not Ast2 would be helpful.

Comment 2

In terms of spatial transcriptomics analysis, the finding and co-localisation of Ast3 astrocytes and Mes2 glioma cells is interesting and relevant.

Comment 3

The addition of the CGGA/TCGA survival analysis is very welcome and strengthens the case for tissue State B as a composite cell state driving prognosis.

Separating out the individual cell components and finding they do not individually correlate with prognosis is relevant but to me the most important analysis here would be to subtract the astrocyte signature from the prognosis calculation.

The link between myeloid/glioma/T-cell is well recognized but the fundamental question here is how glioma associated astrocytes play a part in this process, therefore it would be important in this case to show some evidence of whether the Ast3 glioma associated astrocyte signature is a passenger phenomenon or strongly contributory to the pathogenicity of tissue state B. A “leave one out” analysis for tissue state B might help clarify how strong the contribution of Ast3 cells are to the pathogenic state and survival. If by removing the Ast3 signature one loses the correlation with survival, then that provides

firmer (indirect) evidence that Ast3 cells are important to prognosis. Unless I have misunderstood, this is a different question to whether each individual cell type is contributory.

Comment 4

Finally, the limitations of snRNA seq in capturing myeloid related gene expression should be commented on (Thrupp et al. 2020), all techniques will of course have their tradeoffs, but it could be that myeloid (or other cell type) gene drop out distorts the captured gene signatures that the analyses are based upon.

References

Thrupp N, Sala Frigerio C, Wolfs L, Skene NG, Fattorelli N, Poovathingal S, Fourne Y, Matthews PM, Theys T, Mancuso R, de Strooper B, Fiers M. Single-Nucleus RNA-Seq Is Not Suitable for Detection of Microglial Activation Genes in Humans. *Cell Rep.* 2020 Sep 29;32(13):108189. doi: 10.1016/j.celrep.2020.108189. PMID: 32997994; PMCID: PMC7527779.

Other points:

Typos throughout text (pg 4 line 22), (pg5, line 1)

Please review possible typos in Figure text in 1D (What is "Post-ttt" , please explain). Also in 1G.

Reviewer #2 (Remarks to the Author):

I have no remaining concerns. The authors have significantly improved their manuscript with the addition of additional samples and assays. The impact of the manuscript lies in its identification of prognostic tissue compositions, the elucidation of potential novel therapeutic targets, as well as providing a welcome and timely resource.

Reviewer #3 (Remarks to the Author):

The authors have worked intensively on the criticisms raised and have partially answered them sufficiently. However, it is apparent at first glance that only a fraction of the spatial transcriptomic datasets presented should be used. The quality is significantly lower compared to previously published

data. ST with fewer than 500 UMIs per spot are basically unusable. Additionally, the precise annotation of regions and tumor morphology is difficult to evaluate due to the lack of staining (no H&E). A pure DAPI and NeuN staining is not very informative, why not at least GFAP + IBA1? There are already clearly visible artifacts in the DAPI staining that suggest insufficient preparation when embedding the samples in OCT. Although it is known that the Visium technology and especially the sample preparation can be difficult, this must at least be discussed.

For the analysis, the authors were very creative and, although somewhat cumbersome, were able to substantiate their hypothesis. Since the criticisms that I have raised cannot be easily resolved, I kindly request a sufficient discussion of the quality issues and limitations.

REVIEWER COMMENTS

Reviewer #1 (Remarks to the Author):

The authors have re-submitted their manuscript and should be commended on the fact that many of the points raised have been addressed and the overall story is much more compelling.

Comment 1

The addition of the validation cohort strengthens the contention that distinct astrocyte subgroups exist in these sample cohorts and further defines the Ast 1-3 states. This in turn feeds into the cross validation with neurodegenerative diseases with Ast3 having more similarities to both AD and HD than just HD as previously described.

Perhaps I am mistaken but the link between CLU and Ast3 does not seem particularly strong. If anything CLU expression, which we are using as a surrogate markers for astrocytes associated with neurodegenerative diseases, is higher in Ast2 (see annotated figure below). Indeed with CLU overexpression the increase of Ast3 associated genes such as CHI3L1, ID3 are also highly expressed in Ast2. Some explanation as to why CLU overexpression is felt to be more representative of Ast3 and not Ast2 would be helpful.

Response: We thank the reviewer for providing us the opportunity to clarify, and confirm that CLU overexpressing cells are representative of Ast3, as shown in **Figure 1 Panel F** and **Supplementary Table 4**. We believe that the reviewer's comment refers to **Supplementary Figure 13 Panel B** where we show expression of CLU in unsupervised astrocyte clusters (Louv_1-Louv_3). Notably, while the unsupervised clustering showed similar patterns to our supervised clustering, it did not recapitulate the cluster specific expression of CLU. Rather, it shows CLU is expressed at similar levels in two of the unsupervised clusters (Louv 1 and Louv 3). We have clarified this point in the supplementary results.

To further demonstrate the relationship between CLU and Ast3, we performed GSEA analysis of the astrocyte state signatures (Ast1-3) in the CLU overexpressing astrocytes DEGS (comparing CLU overexpressing astrocytes versus GFP control astrocytes). The results showed that the CLU induced genes are significantly positively enriched in the Ast3 gene signature – **Additional Figure 1**. We have now included these results in **Supplementary Figure 13 Panel I**.

Additional Figure 1 GSEA of Ast3 geneset in CLU-OE vs GFP control astrocyte DEGs

Comment 2

In terms of spatial transcriptomics analysis, the finding and co-localisation of Ast3 astrocytes and Mes2 glioma cells is interesting and relevant.

Response: We thank the reviewer for their encouraging feedback.

Comment 3

The addition of the CGGA/TCGA survival analysis is very welcome and strengthens the case for tissue State B as a composite cell state driving prognosis. Separating out the individual cell components and finding they do not individually correlate with prognosis is relevant but to me the most important analysis here would be to subtract the astrocyte signature from the prognosis calculation.

The link between myeloid/glioma/T-cell is well recognized but the fundamental question here is how glioma associated astrocytes play a part in this process, therefore it would be important in this case to show some evidence of whether the Ast3 glioma associated astrocyte signature is a passenger phenomenon or strongly contributory to the pathogenicity of tissue state B. A “leave one out” analysis for tissue state B might help clarify how strong the contribution of Ast3 cells are to the pathogenic state and survival. If by removing the Ast3 signature one loses the correlation with survival, then that provides firmer (indirect) evidence that Ast3 cells are important to prognosis. Unless I have misunderstood, this is a different question to whether each individual cell type is contributory.

Response: We appreciate the reviewer’s comment. To address this point, we have repeated the survival analyses in **Figure 3**, subtracting the effect of Ast3 by regressing out the contribution of the Ast3 gene signature in the Cox proportional hazard model. The results show that when controlling for the Ast3 gene signature in this way, enrichment of tissue state B is no longer significantly associated with an increased hazard of death (see **Additional Figure 2**). As the reviewer suggests, this provides further evidence that Ast3 is important for prognosis. We have now added this analysis to **Figure 3, Panel H**.

Additional Figure 2 Survival analysis as in figure 3H, correcting for Ast3 signature

Comment 4

Finally, the limitations of snRNA seq in capturing myeloid related gene expression should be commented on (Thrupp et al. 2020), all techniques will of course have their tradeoffs, but it could be that myeloid (or other cell type) gene drop out distorts the captured gene signatures that the analyses are based upon.

Response: We have now updated the discussion section to reflect the limitations of snRNAseq as suggested, have added the following reference:

Thrupp N, Sala Frigerio C, Wolfs L, Skene NG, Fattorelli N, Poovathingal S, Fourné Y, Matthews PM, Theys T, Mancuso R, de Strooper B, Fiers M. Single-Nucleus RNA-Seq Is Not Suitable for Detection of Microglial Activation Genes in Humans. *Cell Rep.* 2020 Sep 29;32(13):108189. doi: 10.1016/j.celrep.2020.108189. PMID: 32997994; PMCID: PMC7527779.

Other points: Typos throughout text (pg 4 line 22), (pg5, line 1)

Please review possible typos in Figure text in 1D (What is “Post-ttt” , please explain). Also in 1G.

Response: We have now corrected these typos.

Reviewer #2 (Remarks to the Author):

I have no remaining concerns. The authors have significantly improved their manuscript with the addition of additional samples and assays. The impact of the manuscript lies in its identification of prognostic tissue compositions, the elucidation of potential novel therapeutic targets, as well as providing a welcome and timely resource.

Response: We thank the reviewer for their input.

Reviewer #3 (Remarks to the Author):

The authors have worked intensively on the criticisms raised and have partially answered them sufficiently. However, it is apparent at first glance that only a fraction of the spatial transcriptomic datasets presented should be used. The quality is significantly lower compared to previously published data. ST with fewer than 500 UMIs per spot are basically unusable. Additionally, the precise annotation of regions and tumor morphology is difficult to evaluate due to the lack of staining (no H&E). A pure DAPI and NeuN staining is not very informative, why not at least GFAP + IBA1? There are already clearly visible artifacts in the DAPI staining that suggest insufficient preparation when embedding the samples in OCT. Although it is known that the Visium technology and especially the sample preparation can be difficult, this must at least be discussed.

For the analysis, the authors were very creative and, although somewhat cumbersome, were able to substantiate their hypothesis. Since the criticisms that I have raised cannot be easily resolved, I kindly request a sufficient discussion of the quality issues and limitations.

Response: We appreciate the reviewer’s feedback and have now updated the discussion section to include a discussion of the quality and limitations of our ST data set.

REVIEWERS' COMMENTS

Reviewer #1 (Remarks to the Author):

none

Reviewer #3 (Remarks to the Author):

No further comments, happy to see the work published